# Localized Curvature-based Combinatorial Subgraph Sampling for Large-scale Graphs

## Abstract

This paper introduces a subgraph sampling method based on curvature to train large-scale graphs via mini-batch training. Owing to the difficulty in sampling globally optimal subgraphs from large graphs, we sample the subgraphs to minimize the distributional metric with combinatorial sampling. In particular, we define a combinatorial metric that distributionally measures the similarity between an original graph and all possible node and edge combinations of the subgraphs. Further, we prove that the subgraphs sampled using the probability model proportional to the discrete Ricci curvature (*i.e.*, Ollivier-Ricci curvatures) of the edges can minimize the proposed metric. Moreover, as accurate calculation of the curvature on a large graph is challenging, we propose to use a localized curvature considering only 3-cycles on the graph, suggesting that this is a sufficiently approximated curvature on a sparse graph. In addition, we show that the probability models of conventional sampling methods are related to coarsely approximated curvatures with no cycles, implying that the curvature is closely related to subgraph sampling. The experimental results confirm the feasibility of integrating the proposed curvature-based sampling method into existing graph neural networks to improve performance.

## 1 Introduction

Most graph data that represent real-world information tend to be on a large scale, with many nodes and edges. However, the training of such large-scale graphs with finite computational resources is challenging, owing to the rapid increase in complexity because of the expanding neighbors. Thus, a large graph must be transformed into smaller subgraphs to facilitate the use of various graph neural models, regardless of the size of the graph. However, inevitable errors are induced by unsampled nodes and edges when the original large-scale graph is divided into independent subgraphs. In general, graph neural models aggregate neighbor node features along with graph structures. Because subgraphs are composed of partial nodes and edges, they have sparse structures that aggregate only a portion of the adjacent nodes.

In this paper, we argue that informative subgraphs can be made similar to the original graph using only partial nodes and edges. Consequently, we present a distributional metric that measures the accuracy of a subgraph when approximating the distribution defined in the original graph. In addition, we present a combinatorial distributional metric that measures the similarity (as an expected value) between all the subgraphs sampled from the probability model and the original graph. This allows us to recognize distributional differences using only sampling probability models, thus eliminating the need for actual subgraph samples.

Submitted to 36th Conference on Neural Information Processing Systems (NeurIPS 2022). Do not distribute.

We find that sampling the edges with large curvatures is equivalent to reducing the proposed distributional difference. Curvature is a geometric property that represents the local structure of a space. In particular, the discrete Ricci curvature (Ollivier-Ricci curvature) can be defined in the graph structure. Thus, many studies have suggested that the structural information of graphs can be employed via the curvature. In context of these studies, we suggest that curvature can also be utilized for subgraph sampling in the course of mini-batch training of large graphs. Consequently, through suitable experiments, we demonstrate that the proposed **Lo**calized **Cur**vature(**LoCur**)-based subgraph sampling method improves the performance of various graphs and graph neural models.

## 2   Related Work

Graph neural networks (GNNs) have been proposed for learning useful information from graph data. However, in the process of aggregating connected adjacent nodes, computational costs tend to increase rapidly with increase in the size of the graph. Thus, the following approaches are introduced to efficiently learn large-scale graphs.

**(Parameters Saving)** The first approach involves reducing parameters of GNNs. In general, the accuracy of GNNs is not linearly proportional to the size of parameters. Thus, in [39], the linear activated swallow GNN learned the feature space that approximates deep non-linear activated GNNs. In [11], the parameters of the GNN model were parallelized. However, because these methods are based on specific models, they cannot be easily combined with various GNNs.

**(Computation Saving)** The second approach involves dynamic selection of the nodes to be considered for each layer. To realize this, the neighbor sampling method with historical activation was proposed [4], and the variance was reduced via neighbor sampling-based training. Further, the iid node importance sampling by Monte-Carlo approach was proposed to prevent recursive neighborhood expansion [5]. These methods approximated the mini-batch gradients. In [17], the conditional probability model for each layer was learned, and the nodes computed in the next layer were sampled according to the probabilities. However, these methods are not model-free methods.

**(Input Saving)** The third approach involves sampling relatively small subgraphs as inputs to train networks. The subgraph sampling methods include graph coarsening, graph partitioning, and graph covering. For example, graphs were filtered through spectral coarsening [33, 9], and adaptive edge masks were used to drop connections [15]. However, these methods sampled the edges to render sparse graphs [43], wherein the number of nodes were not reduced but only edges were removed. In [18, 3], linked nodes were merged into a hyper node to reduce the size of the subgraph. However, this method encountered difficulties in solving node classification problems, which needs to classify each node. In contrast, the following method obtained a subgraph using only structure information without node features. These methods are applicable to large graphs because they sample the subgraphs via combinatorial node and edge samplings. In [14], a subgraph was obtained by sampling the neighborhoods of the seed nodes. In [45], subgraph sampling methods were proposed based on node degrees, edge weights, and random walks. In [7], a subset of clusters was used as a subgraph through graph clustering. In [1], personalized page rank scores were used to sample nodes in a subgraph.

**(Curvature on Graphs)** The graph-structured data is a dependent object, in which nodes are connected along edges. In addition, the curvature is defined based on structural information. The Ollivier-Ricci curvature was proposed in [28] considering random-walk-based probability measures with the Markov chain and the 1-Wasserstein transportation distance to compute the distance between the probability measures. Another method of capturing the topological structure of the graph is presented in [10] and compared in [31]. The curvature on graphs has been studied in many applications such as finding communities [26], curvature attentive networks [42, 40, 38], over-squashing alleviation [36], preventing distortion of graph embedding [29], and generating graphs [23]. In previous studies, the results have shown the existence of a significant relationship between the curvature of the graph and the informative geometric properties. In this paper, we present that a curvature-based input saving approach helps to approximate the original graph to a sparse subgraph.

## 3 Metric for Combinatorial Subgraphs

Let $\mathcal{G} = (\mathcal{V}, \mathcal{E})$ be a graph with the combinatorial structure of nodes $\mathcal{V}$ and edges $\mathcal{E}$. We can sample subgraphs $\widehat{\mathcal{G}} = (\widehat{\mathcal{V}}, \widehat{\mathcal{E}})$ with sizes $|\widehat{\mathcal{V}}| \ll |\mathcal{V}|$ from $\mathcal{G}$. However, numerous subgraphs can be obtained from the graph $\mathcal{G}$. Therefore, the criteria for determining good subgraphs must be defined according to the purpose of sampling. Please note that the subgraphs are sampled to train a large-scale graph $\mathcal{G}$ by dividing them into mini-batches $\left\{ \widehat{\mathcal{G}}_i \right\}$, because training the large graph at once is difficult. Therefore, for unbiased training through mini-batches, the subgraphs should be sampled to be similar to the original graph. In graph theory, the traditional metrics for the similarity between two different graphs can be defined in various manner.

**(Conventional Metric)** The graph edit distance ($GED$) is a measure of matching two different graphs by calculating the editing cost to obtain one graph from another, *i.e.*, $d_{GED}(\mathcal{G}, \widehat{\mathcal{G}}) = \min_{(e_1, e_2, ..., e_k) \in E(\mathcal{G}, \widehat{\mathcal{G}})} \sum_{i=1}^{k} c(e_i)$, where $c(e_i)$ denotes the cost of each edit operation, including vertex insertion, vertex deletion, edge insertion, and edge deletion. $E(\mathcal{G}, \widehat{\mathcal{G}})$ denotes the set of edit paths. A single path comprises several edit operations $(e_1, e_2, ..., e_k)$ used to modify $\widehat{\mathcal{G}}$ to match another graph $\mathcal{G}$. In contrast to the exact graph matching problem that solves graph isomorphism, the graph edit distance measures the similarity between two different graphs (*i.e.*, error-tolerant graph matching). Thus, it calculates the minimum error in matching one graph to another. The linear $p$-norm can also be defined as $d_A(\mathcal{G}, \widehat{\mathcal{G}}) = \|A - \widehat{A}\|_p$, where the $A$ and $\widehat{A}$ denote the adjacency matrices of $\mathcal{G}$ and $\widehat{\mathcal{G}}$, respectively. However, if the $GED$ measures the distance $d_{GED}(\mathcal{G}, \widehat{\mathcal{G}})$ between the original graph and subgraph, only one edit path is obtained. This is similar to the addition of the cost of the number of nodes to the linear distance, $d_A(\mathcal{G}, \widehat{\mathcal{G}})$. Therefore, the metric $GED$ and the linear $p$-norm do not reflect the structure of the graph. In contrast, the spectral norm measures the norm of the eigenvalues, $d_\lambda(\mathcal{G}, \widehat{\mathcal{G}}) = \sum_{i=1 \cdots n} \|\lambda_i - \widehat{\lambda}_i\|$. However, it requires eigendecomposition of the original graph and subgraph, the application of which to large graphs is challenging.

**(Distributional Metric)** To overcome the problems of conventional metrics, we define the distributional metric, *i.e.*, the distance between the probability measures defined on the unweighted graph. Let $(\mathcal{V}, d)$ be a metric space on the connected graph, where $d(x, y)$ is the distance between the nodes $x$ and $y$. We define the probability measures for each node to represent the local structure around the nodes. Then, the metric $d_m(\mathcal{G}, \widehat{\mathcal{G}})$ between the original graph $\mathcal{G}$ and the subgraph $\widehat{\mathcal{G}}$ is defined.

**Definition 1 (Metric for Deterministic Subgraph)** *Let $\mu$ and $\widehat{\mu}$ be probability measures for original and sampled subgraphs, respectively. $d(x, y)$ is the distance of the geodesic path between nodes $x$ and $y$, and $\Pi(\mu, \widehat{\mu})$ denotes the set of all couplings between two measures, $\mu$ and $\widehat{\mu}$. Then, the distance $d_m(\mathcal{G}, \widehat{\mathcal{G}})$ is defined as the distributional distance between $\mu$ and $\widehat{\mu}$.*

$$d_m(\mathcal{G}, \widehat{\mathcal{G}}) = \inf_{\rho \in \Pi(\mu, \widehat{\mu})} \int_{x, y \in \mathcal{V} \times \mathcal{V}} d(x, y) \rho(x, y). \tag{1}$$

Using Definition 1, we can represent the distance between the original graph and the deterministic subgraph. However, because various subgraphs can be obtained from one graph, we extend the metric presented in Definition 1 to the metric for non-deterministic subgraphs, as follows:

**Definition 2 (Metric for Non-deterministic Subgraph)** *A subgraph can be considered as a set of sampled nodes and edges derived from combinatorial sampling. Therefore, the distributional metric can be redefined as the combination of node-wise metrics or edge-wise metrics. Then, the probability measure of the non-deterministic subgraph is represented as the expected value of node-wise measures, $\sum_{x \in \mathcal{V}} p(x) \mu_x$, where $\mu_x$ is a probability measure for the node $x$. Here, $\mu_x$ can be decomposed as edge-wise measures along the edges, $\sum_{y \in \mathcal{N}(x)} p(y|x) \delta_y$, where $p(y|x)$ is the probability that $y$ is sampled to be connected to $x$, and $\delta_y = 1_y$. We can define the combinatorial metric between the original graph and the probabilistic sampled subgraph, as the upper bound of 1.*

$$d_m(\mathcal{G}, \widehat{\mathcal{G}}) \leq \sum_{x \in \mathcal{V}} p(x) d_m(\mathcal{G}_x, \widehat{\mathcal{G}}_x) \leq \sum_{x \in \mathcal{V}} p(x) \left( \sum_{y \in \mathcal{N}(x)} p(y|x) d_m(\mathcal{G}_x, v_y) \right), \tag{2}$$

where the metric $d_m(\mathcal{G}, \widehat{\mathcal{G}})$ is upperbounded by the weighted average of the distances between the local structural graph $\mathcal{G}_x$ and each node $v_y$. Although the weighted sum of distances for components is the upper bound, this method enables the determination of the better subgraph sampling model, using only probability models instead of actual subgraph samples. For example, a random edge sampler samples the subgraphs with an approximation of $\sum_{x \in \mathcal{V}} \frac{1}{|\mathcal{V}|} \sum_{y \in \mathcal{N}(x)} \frac{1}{|\mathcal{N}(x)|} d_m(\mathcal{G}_x, v_y)$, because it is a uniform sampling. However, if a sampler has a sampling probability model $p(x)p(y|x) \propto \frac{1}{d_m(\mathcal{G}_x, v_y)}$ that is inversely proportional to the distributional difference of each edge, it can be considered a better subgraph sampler because it more probable to obtain subgraphs that approximate the original graph.

# 4  Curvature-based Local Structural Approximation

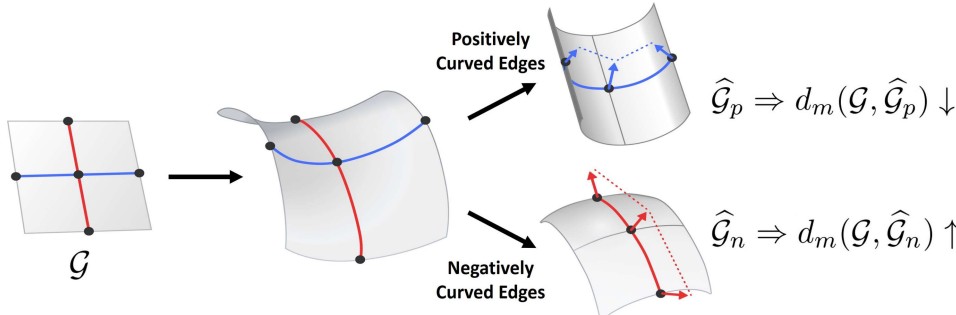

Figure 1: **Curvature-based distributional difference.** If the graph $\mathcal{G}$ is not in the form of a regular grid, its edges have different distributional distances depending on the structure of the graph. Then, the subgraph $\widehat{\mathcal{G}}_p$ consisting of positively curved edges (blue) are more similar to the original graph than the subgraph $\widehat{\mathcal{G}}_n$ consisting of negatively curved edges (red).

We introduce the discrete Ricci curvature $\kappa_{xy}$ in (4), which is defined as the ratio of the distance of probability measures to the distance of nodes. We show that subgraph sampling that maximizes the curvature amounts to minimizing the distributional difference between the original graph and subgraph. Intuitively, $d_m$ is determined by the local structure around two nodes, because it can be defined as the cost of moving from adjacent nodes of one node to adjacent nodes of another node. Thus, subgraph sampling to maximize the curvature enables local structural approximation because it minimizes the distribution metric between the original graph and the subgraph.

**(Ollivier-Ricci Curvature)** Let $\mathcal{G} = (\mathcal{V}, \mathcal{E})$ be a graph with nodes $\mathcal{V}$ and edges $\mathcal{E}$. $\mathcal{N}(x)$ denotes adjacent nodes of node $x \in \mathcal{V}$ and $\mathsf{d}_x$ is the degree of node $x$. Assume that $m_x$ is the probability measure of node $x$ by the random walk kernel $m$ along the Markov chain. Then, the metric measure space can be defined as $(\mathcal{V}, d, m)$. The discrete probability measure $m_x$ for each node $x \in \mathcal{V}$ is defined as follows.

$$m_x(v) := \begin{cases} \mathbf{1}_x(v) & \text{if } \mathsf{d}_x = 0, \\ \frac{1}{\mathsf{d}_x} \cdot \mathbf{1}_{\mathcal{N}(x)}(v) & \text{if } \mathsf{d}_x > 0. \end{cases} \tag{3}$$

With the probability measures $m_x, m_y$ for nodes $x, y \in \mathcal{V}$ of the graph, the transport plan between the two probabilities, $m_x(\mathcal{V})$ and $m_y(\mathcal{V})$, is represented as the coupling measure $\gamma$, where $\Pi(m_x, m_y)$ is the set of measures projecting $m_x$ to $m_y$. Distance $d(x, y)$ is defined as the length of the shortest path between nodes $x, y$ for the 1-geodesic distance. This is the optimal transport problem between sets of nodes $\mathcal{N}(x)$ and $\mathcal{N}(y)$. Then, the 1-Wasserstein distance $\mathcal{W}_1$ is defined as the minimal cost for transportation between two measures, *i.e.*, $\mathcal{W}_1(m_x, m_y) = \inf_{\gamma \in \Pi(m_x, m_y)} \int_{u,v \in \mathcal{V} \times \mathcal{V}} d(u,v) \mathrm{d}\gamma(u,v)$.

If there is at least one path for nodes $x, y$, we can define the Ollivier-Ricci curvature $\kappa_{xy}$ of the edge $(x, y)$ in the metric measure space $(\mathcal{V}, d, m)$, as follows:

$$\kappa_{xy} = 1 - \frac{\mathcal{W}_1(m_x, m_y)}{d(x, y)}. \tag{4}$$

(**Local Structural Approximation**) The Ollivier-Ricci curvature is defined on the metric measure space $(\mathcal{V}, d, m)$ for graphs, in which the large curvature $\kappa_{xy}$ indicates the small distance $\mathcal{W}_1(m_x, m_y)$ between the probability measures $m_x, m_y$ for two nodes $x, y$. Because the probability measure is defined through a random-walk kernel, the distribution is distributed uniformly to 1-hop neighboring nodes in the unweighted graph. Then it effectively represents the local structural information. Therefore, the proposed subgraph sampling proportional to the curvature reduces the distributional difference $d_m(\mathcal{G}_x, \widehat{\mathcal{G}}_x)$, while also approximating the local structure of the graph. As shown in Fig.3, even if the same number of edges are sampled equally based on the node, subgraph $\widehat{\mathcal{G}}_p$ with positively curved edges can be more distributionally and structurally similar to the original graph than subgraph $\widehat{\mathcal{G}}_n$ with negatively curved edges.

**Proposition 1** *Let $(\mathcal{V}, d, m)$ be the metric space with the random walk $m$. Then, $m_x$ is the probability measure defined in 3 for the local structure $\mathcal{G}_x$, and $w_x$ is the probability measure defined as $w_x = \sum_{y \in \mathcal{V}} p(y|x) \cdot \delta_y$, where $\sum_{y \in \mathcal{N}(x)} p(y|x) = 1$ and $\delta_y = 1_y$, for the local structure $\widehat{\mathcal{G}}_x$. The distributional difference of probabilistic sampled subgraphs is bounded by curvatures, as follows:*

$$d_m(\mathcal{G}_x, \widehat{\mathcal{G}}_x) \leq 2 - \sum_{y \in \mathcal{N}(x)} p(y|x)\kappa_{xy}, \tag{5}$$

where $\kappa_{xy}$ denotes the Ollivier-Ricci curvature of edge $(x, y)$. Proposition 1 shows that the distributional difference around node $x$ of the probabilistic sampled subgraphs is inversely proportional to the expected value of the curvatures of the edges linked to $x$. This implies that the distributional difference between the sampled subgraph and original graph decreases with the increase in the expected value of the curvature; that is, the probability model, which frequently samples edges with large curvatures, can obtain more structurally approximated subgraphs. Please note that the upper bound of the distributional difference in Proposition 1 can be extended to n-hop to capture a wider scope of local structure near the node. Thus, we also present the upper bound of the distributional difference between the n-hop diffused probability measure $m_x^{*n}$ and the probability measure $w_x$ of the probabilistic sampled subgraphs, as follows:

**Corollary 1.1** *$m^{*n}$ denotes the probability measure of n-step random walks on a Markov chain. It can be considered as the state of n-hop propagation in the original graph. Suppose that the graph $\mathcal{G}_p = (\mathcal{V}, \mathcal{E})$ is composed of positively curved edges such as $\kappa_{xy} \geq \kappa > 0$ for any pair of nodes $x, y$ in $\mathcal{V}$. Then, the upper bound of distributional difference between $m_x^{*n}$ and $w_x$ is bounded as follows.*

$$d_m(\mathcal{G}_x^{*n}, \widehat{\mathcal{G}}_x) \leq \frac{(1 - \kappa) - (1 - \kappa)^n}{\kappa} + d_m(\mathcal{G}_x, \widehat{\mathcal{G}}_x). \tag{6}$$

Corollary 1.1 shows that the distributional difference between the n-hop local structure $\mathcal{G}_x^{*n}$ and sampled local structure $\widehat{\mathcal{G}}_x$ is also bounded in relation to the 1-hop local distributional difference in Proposition 1. Subsequently, we suggest that subgraphs consisting of edges with large curvatures have generalization effects, which can be advantageous for learning the model.

**Proposition 2** *Let $\mathcal{P}_i = (x_i = v_0, v_2, ..., v_n = y_i)$ and $\mathcal{P}_j = (x_j = u_0, u_1, ..., u_n = y_j)$ be the geodesic paths for any shortest path of $(x_i, y_i)$ and $(x_j, y_j)$, respectively. The distance for each edge is defined as $d(v_i, v_{i+1}) = d(u_i, u_{i+1})$. Then, if the curvatures of the paths $\mathcal{P}_i, \mathcal{P}_j$ satisfy the inequalities $\kappa_{v_i v_{i+1}} > \kappa_{u_i u_{i+1}}$ for any edge $(v_i, v_{i+1})$ and $(u_i, u_{i+1})$ in the paths, the following inequality holds for the gradients between starting and ending nodes in the paths:*

$$\nabla_{x_i y_i \in \mathcal{P}_i}(f + \Delta f) \leq \nabla_{x_j y_j \in \mathcal{P}_j}(f + \Delta f) \tag{7}$$

*for any 1-Lipschitz function $f$.*

Proposition 2 implies that the two paths have the same length but can yield different gradients. The gradient is proportional to the sum of the edge curvatures in the path. Therefore, if it is possible to walk along the edges with large curvatures, the distance of node features in the starting and ending nodes of the path can be close, even for the same length of walks. Thus, propagation by edges with

a large curvature can smoothen the feature space. And this can give a generalization effect. The locally smoothed feature space can be defined as $\|f_x - f_y\| < \alpha$ for any pair of nodes $(x, y)$ with $d(x, y) \leq n$. The small gradient $\nabla_{xy}$ in the n-bound indicates that smooth feature space is obtained even with relatively distant nodes. In Section 3, we conduct experiments on the generalization effect to demonstrate the accuracy with which all nodes can be predicted through learning with only small partial nodes.

## 5   Localized Curvature-based Combinatorial Sampling

Subgraph sampling is used to sample mini-batches for a large-scale graph. However, sampling the optimal subgraphs by considering the whole structure of the large graph is challenging. Therefore, the previous section shows that local structural approximation of subgraph using curvature is possible instead. However, it is still difficult to calculate the exact curvatures of the edges due to the high computational cost of solving the optimal transport problem. Thus, we obtain subgraphs through combinatorial sampling proportional to the approximated curvatures of the Definition 3 using the localized structure of the graph with 3-cycles. We show that the approximated curvatures exhibit sufficiently small errors in local structures of sparse graphs.

**Definition 3** ([19], **Localized Curvature with 3-cycles**) $d_x$ and $d_y$ denote the degree of nodes $x$ and $y$, respectively. $\triangle_\sharp(x, y)$ represents the number of triangles (3-cycles) including the edge $(x, y)$. The operations $d_x \wedge d_y$ and $d_x \vee d_y$ denote $\min[d_x, d_y]$ and $\max[d_x, d_y]$, respectively. $(\cdot)_+$ is defined as $\max[\cdot, 0]$. Then, the localized curvature $\kappa_{xy}$ for the edge $(x, y)$ has the following lower bound.

$$\kappa_{xy} \geq -\left(1 - \frac{1}{d_x} - \frac{1}{d_y} - \frac{\triangle_\sharp(x,y)}{d_x \wedge d_y}\right)_+ - \left(1 - \frac{1}{d_x} - \frac{1}{d_y} - \frac{\triangle_\sharp(x,y)}{d_x \vee d_y}\right)_+ + \frac{\triangle_\sharp(x,y)}{d_x \vee d_y}. \quad (8)$$

The lower bound of $\kappa_{xy}$ in (8) can be considered the localized curvature for ((4) and Fig.2A) in the 1-hop local structures $\mathcal{N}(x) \cup \mathcal{N}(y)$. As this localized curvature only considers adjacent neighbors, it is not accurate. However, the localized curvature with 3-cycles ((8) and Fig.2B) is within the tolerance range of the exact curvature for sparse graphs. We also present

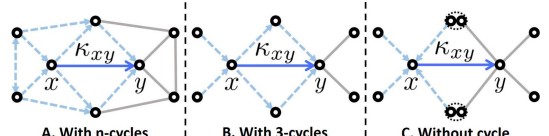

A. With n-cycles    B. With 3-cycles    C. Without cycle

Figure 2: **Approximated curvatures in the local structure.**

that other conventional subgraph sampling methods are related to more approximated curvatures with no cycles ((9) and Fig.2C), and argue that curvature is closely related to subgraph sampling. It assumes that there are no cycle $\triangle_\sharp(x, y) = 0$ in the local structure of the graph even if there are cycles. Therefore, the lower bound of the curvature without cycles [24] is defined as follows.

$$\kappa_{xy} \geq -2\left(1 - \frac{1}{d_x} - \frac{1}{d_y}\right)_+, \quad (9)$$

where $d_x$ and $d_y$ denote the degrees of nodes $x, y$ of edge $(x, y)$, respectively. Conventional combinatorial subgraph sampling methods are related to this localized curvature without a cycle, as Table 1.

Table 1: Conventional methods can be interpreted as sampling related to the coarse curvature.

| Sampler | Probability model | With curvature |
|---------|-------------------|----------------|
| Edge[45] | $p(e_{xy}) \propto \frac{1}{d_x} + \frac{1}{d_y}$ | $p(e_{xy}) \propto \frac{\kappa_{xy}+2}{2}$ |
| Node[45] | $p(x) \propto (\sum_{y \sim x} \frac{1}{d_x} + \frac{1}{d_y} - 1)^2$ | $p(x) \propto (\sum_{y \sim x} \frac{\kappa_{xy}+2}{2} - 1)^2$ |
| Cluster[7] | $p(x) \propto c(x)$ | $p(x) \propto \frac{\sum_{y \sim x}(\kappa_{xy}+2)-4}{3d-1}$ |

We show that the local structure $\mathcal{N}(x) \cup \mathcal{N}(y)$ of the graph does not have the shortest path beyond 5-cycles, thus the localized curvature with 3-cycles has very small errors in the sparse large graph.

**Proposition 3 (Error bound of localized curvature)** *Let $\mathcal{G}(n,p)$ be the ER-graph (Erdős-Rényi model) with $n > 4$ nodes and edges connected by the probability $0 \leq p \leq 1$. Then, the degrees $\mathsf{d}_x, \mathsf{d}_y$ of nodes $x, y$ are expected as $\mathsf{d}_x = \mathsf{d}_y = \mathsf{d} = (n-1)p$. Let $\triangle_\sharp(x,y)$ be the number of 3-cycles, $\square_\sharp(x,y)$ be the number of 4-cycles, $\pentagon_\sharp(x,y)$ be the number of 5-cycles. In the local structure, the shortest distance between nodes can be obtained by considering only 5-cycles or less, so there is a small error in the approximated curvature $\widehat{\kappa}_{xy}$ with 3-cycles for sparse large graphs:*

$$\|\kappa_{xy} - \widehat{\kappa}_{xy}\| \leq \frac{2}{\mathsf{d}}\square_\sharp(x,y) + \frac{1}{\mathsf{d}}\pentagon_\sharp(x,y) \leq 2\mathsf{d}p + 2\mathsf{d}^2p \tag{10}$$

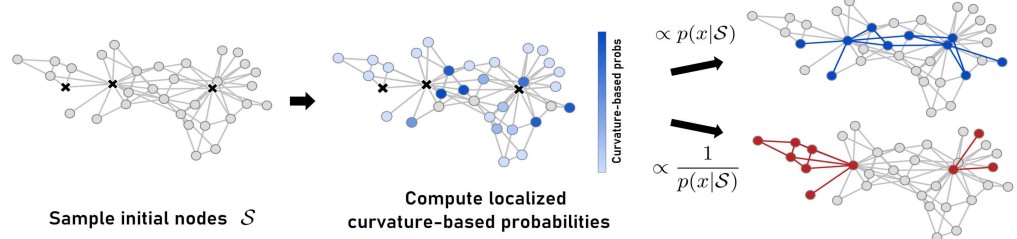

Figure 3: **Curvature-based subgraph sampling.** First, we sample the initial nodes $\mathcal{S}$, then the probabilities $p(x|\mathcal{S})$ is computed based on the curvatures. With the proposed method, we can sample nodes (blue) to maximize the curvatures in a subgraph.

**(Proposed Method)** We propose a localized curvature-based sampling method to sample the subgraphs from large graphs. There are studies [8, 30, 34, 2] that solve optimization problems to sample the subgraphs. However, because they have difficulty in dealing with large graphs, we present a combinatorial approach that samples nodes and edges as like [14, 45, 7, 1]. Furthermore, while conventional methods have been described through variance reduction or node degree, we show that they can be interpreted by using the coarse curvature (9) in Table 1. Therefore, we propose the algorithm that locally approximates the optimal subgraph by finding the subgraph with large curvatures in Definition 3.

The curvature $\kappa_{xy}$ is defined for edges $(x,y)$, in which the sampling probability for each edge is defined as $p(y|x) \propto \kappa_{xy}$ in Definition 2. However, because $p(y|x)$ is bounded in the local structure, it can be defined via normalization, *i.e.*, $p(y|x) = \frac{\kappa_{xy}}{\sum_{v \in \mathcal{N}(x)} \kappa_{xv}}$. Alternatively, the probability for each edge can be defined directly to the structural normalized curvature, *i.e.*, $p(y|x) \propto \frac{\kappa_{xy}}{\mathsf{d}_y}$.

Using the above probability model, we extend local optimal subgraphs instead of finding a global optimal subgraph. In the beginning, there are no sampled node, so initial nodes $x_0 \in \mathcal{S}_0$ are sampled by $p(x_0|\mathcal{V}) \propto \sum_{y \in \mathcal{N}(x_0), y \in \mathcal{V}} p(x_0|y)$. After sampling the initial nodes, linked nodes $x_{i+1} \in \mathcal{N}(\mathcal{S}_i)$ have sampling probabilities $p(x_{i+1}|\mathcal{S}_i) \propto \sum_{y \in \mathcal{N}(x_{i+1}), y \in \mathcal{S}_i} p(x_{i+1}|y)$. Our method starts at the initial nodes $\mathcal{S}_0$, then extends to the connected nodes $\mathcal{S}_1$ to maximize the sum of curvatures of edges in the subgraph $\widehat{\mathcal{G}} = \mathcal{S}_0 \cup \mathcal{S}_1 \cup ... \cup \mathcal{S}_t$. If we sample $n$ subgraphs with the size of $m$ in $t$ steps, the whole pipeline of the proposed method is in Algorithm 1. Further details can be found in Appendix E.

---

**Algorithm 1** Localized Curvature-Based Sampling (**LoCur**)

---

**Inputs:** $\mathcal{G} = (\mathcal{V}, \mathcal{E})$, iters $n$, sample size $m$, steps $t$ / **Output:** sampled subgraphs $[\widehat{\mathcal{G}}_i]_{k=1}^n$

---

$p(x|\mathcal{V}) \propto \sum_{y \sim x, y \in \mathcal{V}} p(x|y)$        $\triangleright$ compute localized curvature-based probabilities
**for** $k = 1$ **to** $n$ **do**
    $\mathcal{S}_0 = \{v_0, v_1, ..., v_{[m/s]}\} \sim p(x|\mathcal{V})$        $\triangleright$ sample initial nodes
    **for** $i = 0$ **to** $t - 1$ **do**
        $p(x|\mathcal{S}_i) \propto \sum_{y \in \mathcal{N}(x), y \in \mathcal{S}_i} p(x|y)$        $\triangleright$ compute localized curvature-based probabilities
        $\mathcal{S}_{i+1} = \mathcal{S}_i \cup \{u_0, u_1, ..., u_{[m/s]}\} \sim p(x|\mathcal{S}_i)$        $\triangleright$ sample extended nodes
    **end for**
    $\widehat{\mathcal{G}}_k = \mathcal{S}_t$
**end for**

---

## 6 Experiments

Our method proposes to sample the subgraphs that maximize the curvature more directly compared to conventional methods. Therefore, although our method cannot calculate the exact Ollivier-Ricci curvature for the large-scale graphs, it can present the exact curvature calculated by [27] for small graphs (Cora graph [25] with 2708 nodes and 5429 edges, Citeseer graph [12] with 3312 nodes and 4732 edges, and the Pubmed graph [32] with 19717 nodes and 44338 edges). The following experiment in Table 2 was evaluated for 1000 subgraphs with about 10% (200/2708, 300/3327, 2000/19717) partial nodes.

Table 2: **Comparison of Ollivier-Ricci curvature using the Cora, Citeseer, Pubmed graphs.**

| Ollivier-Ricci Curvature | random | neighbor | node | edge | random walk | multi clusters | PPR | LoCur (Ours) | original graph |
|---|---|---|---|---|---|---|---|---|---|
| Cora | -0.08 | -0.06 | -0.09 | -0.08 | -0.07 | -0.02 | -0.08 | **+0.00** | -0.07 |
| Citeseer | -0.05 | -0.06 | -0.09 | -0.06 | -0.06 | -0.01 | -0.07 | **+0.02** | -0.06 |
| Pubmed | -0.08 | -0.07 | **-0.05** | -0.07 | -0.07 | -0.07 | -0.08 | **-0.05** | -0.08 |

We also show sampled subgraphs for the karate club graph [44] with 34 nodes and 78 edges, which is a very small graph, and present results for one of the REDDIT-BINARY graphs [13] with 400 nodes and 460 edges. With this experiment, we can examine the structure sampled by the subgraph. As shown in Fig.4, our method samples the subgraphs to include representative structures without bias compared to other methods.

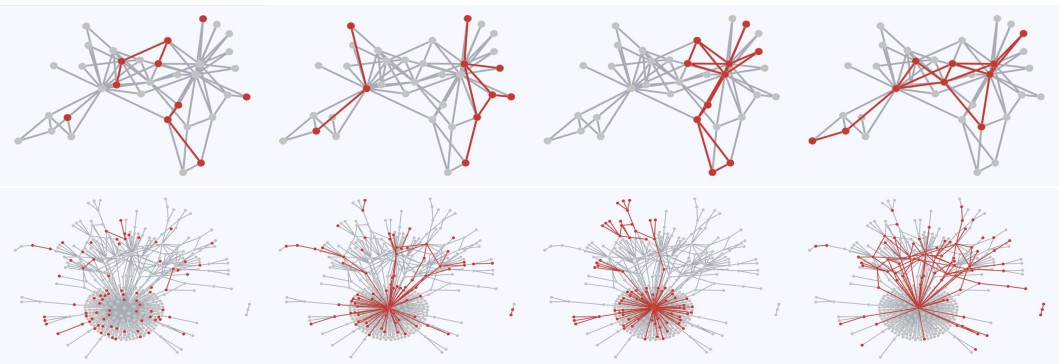

Figure 4: **Subgraph examples for Karate club and REDDIT-BINARY graphs.** This figure show the subgraphs of random, edge, cluster, and LoCur(ours) sampling from left to right. Additional results can be found in Appendix F.

These sampled subgraphs are typically evaluated by the tasks such as node classification and graph classification. However, in the case of semi-supervised node classification, evaluation can be affected by train, valid, and test ratios of nodes within sampled subgraphs. Therefore, we present a labeling task that evaluates the entire nodes (100%) by learning only one sampled subgraph of the specific proportions (1%, 5%, 10%) in 100 experiments. Table 3 shows that our method induces the generalization effect, which yields good performance even with very few nodes.

Our subgraph sampling method can train large-scale graphs, such as mini-batch training, by sampling approximated subgraphs, which are used only for training and the entire graph is used for evaluation. Because each graph has a different size, the size of subgraphs are different for each graph. In addition, all experiments can be reproduced using a common random seed. The sampling costs are negligible because the subgraphs are shuffled and reused to configure the batches following the initial sampling. The proposed method was evaluated based on the highest valid accuracy without any methods, such as early stop or regularization. Consequently, the average accuracy and standard deviation of the training results were obtained for 100 subgraphs by performing the experiment 10 times.

Table 3: **Comparison of labeling task using the ogbn-arxiv, ogbn-mag graphs.**

| Labeling | arxiv-1 (%) | arxiv-5 (%) | arxiv-10 (%) | mag-1 (%) | mag-5 (%) | mag-10 (%) |
|---|---|---|---|---|---|---|
| random | 57.02±2.55 | 65.78±0.71 | 68.16±0.31 | 25.93±0.49 | 29.25±0.21 | 30.94±0.15 |
| neighbor | 58.13±1.37 | 66.28±0.44 | 68.73±0.30 | 24.95±0.45 | 30.42±0.18 | 32.66±0.13 |
| node | 58.55±1.35 | 67.12±0.39 | 69.51±0.23 | 25.03±0.45 | 30.42±0.16 | 32.30±0.11 |
| edge | 60.42±0.88 | 67.10±0.33 | 69.31±0.22 | 25.67±0.41 | 30.55±0.18 | 32.47±0.13 |
| random walk | 59.37±0.84 | 66.66±0.33 | 68.90±0.23 | 25.49±0.35 | 30.47±0.16 | 32.46±0.13 |
| cluster | 53.09±1.80 | 64.54±0.54 | 67.63±0.33 | 22.03±0.65 | 29.14±0.22 | 31.78±0.19 |
| ppr | 56.74±3.46 | 66.13±0.78 | 68.35±0.43 | **26.06±0.36** | 30.41±0.20 | 32.54±0.14 |
| LoCur (Ours) | **62.01±0.72** | **68.09±0.29** | **70.14±0.19** | 25.69±0.48 | **30.78±0.22** | **32.86±0.15** |

The ogbn-arxiv graph [16] in Table 4 comprises $169,343$ nodes and $1,166,243$ edges, which is a directed graph representing citations. We trained the graph neural models with undirected normalization and sampled 100 subgraphs with the size of $1,700(1\%)$ to construct mini-batches. Because our sampling method uses only the structural information of the graph, it does not directly improves the accuracy of the node classification task. Nevertheless, our method produced competitive results.

The ogbn-mag graph [16] in Table 4 consists of $736,389$ nodes and $10,792,672$ edges. We sampled 100 subgraphs with the size of $7,000(1\%)$ to construct mini-batches, and performed the mini-batch training using the sampled subgraphs. The evaluation of the compared methods was conducted using the entire graph. In the case of the GCNII method and ogbn-mag dataset, inferring the original graph is difficult; thus, the results are not available. The accuracy of the sampling methods depends on the characteristics of the graph because subgraph sampling methods use only the graph structural information without feature information. In the case of the ogbn-mag graph, aggregating the feature points of nearby nodes, for example, clustering, induced good results.

Tables 3 and 4, and additional results [1] show that the proposed method produced competitive results for various graphs and GNNs. Our method can efficiently train GNNs using only subgraphs with partial nodes (1%) compared to the GNNs trained for the entire graph.

Table 4: **Comparison of node classification using the ogbn-arxiv and ogbn-mag graphs.**

| Node Classification | arxiv/GCN | | arxiv/GraphSAGE | | mag/GCN | | mag/GraphSAGE | |
|---|---|---|---|---|---|---|---|---|
| | Valid acc (%) | Test acc (%) | Valid acc (%) | Test acc (%) | Valid acc (%) | Test acc (%) | Valid acc (%) | Test acc (%) |
| random | 65.60±1.21 | 64.43±2.06 | 67.15±0.35 | 66.54±0.63 | 28.79±0.80 | 30.02±0.98 | 30.89±0.53 | 31.87±0.47 |
| neighbor | 70.49±0.22 | 69.00±0.59 | 71.12±0.16 | 70.11±0.23 | 32.60±0.62 | 33.14±0.80 | 32.83±0.30 | 33.64±0.37 |
| node | 68.44±0.24 | 67.56±0.37 | 65.32±0.15 | 64.20±0.25 | 28.15±0.42 | 29.35±0.49 | 26.72±0.53 | 28.06±0.57 |
| edge | 69.97±0.27 | 69.12±0.61 | 70.53±0.19 | 69.23±0.30 | 30.83±0.46 | 31.41±0.46 | 31.70±0.32 | 32.66±0.41 |
| random walk | 70.39±0.19 | 69.41±0.29 | 70.78±0.17 | 69.55±0.26 | 31.86±0.18 | 32.48±0.29 | 32.08±0.47 | 32.89±0.48 |
| cluster | 70.58±0.35 | 69.35±0.62 | 70.56±0.19 | 69.09±0.39 | 33.93±0.37 | 34.82±0.39 | 34.41±0.48 | 35.09±0.51 |
| ppr | 67.70±0.55 | 66.45±0.83 | 71.10±0.23 | 70.04±0.62 | 29.67±0.57 | 30.85±0.68 | 31.31±0.52 | 32.04±0.63 |
| **LoCur (Ours)** | **70.91±0.24** | **69.53±0.57** | **71.29±0.15** | **70.15±0.31** | **34.54±0.64** | **35.08±0.77** | **34.69±0.65** | **35.32±0.86** |
| original graph | 73.15±0.08 | 71.97±0.32 | 73.08±0.21 | 71.65±0.46 | 36.54±0.18 | 36.74±0.26 | 36.55±0.40 | 36.69±0.35 |

# 7  Conclusion

To learn a large graph by dividing it into subgraphs, the subgraphs should be sampled to represent the original graph. While finding substructures using curvature in large graphs has been widely studied, we propose that the coarse Ollivier-Ricci curvature bounded by the local structure is closely related to conventional combinatorial subgraph sampling methods. Therefore, our method performs a local optimization search to maximize the curvature of the subgraph based on the localized curvature. Consequently, our method can obtain the combinatorial subgraphs to approximate the original graph by the distribution metric. The experimental results show that subgraphs with the edges of large curvature are suitable for representing the original graph.

---

[1]Please refer to Appendix F for additional experiments.

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
