# A Graph Measures for Subgraph

In Section 3 of the main paper, we compare conventional metrics with distributional metrics. Moreover, we show that distributional metrics are easily decomposed into combinatorial metrics for combinatorial subgraph sampling and they are closely related to subgraph sampling methods proportional to the curvature. In this section, we provide adjuncts on several conventional metrics.

**(Conductance)** The conductance is used as a measure to present the ratio of outer edges in graph theory. Let $\mathcal{S} \subseteq \mathcal{V}$ be sampled partial nodes in the subgraph $\widehat{\mathcal{G}}$. Then, the conductance of the cut $(\mathcal{S}, \mathcal{V}\backslash\mathcal{S})$ in the graph $\mathcal{G} = (\mathcal{V}, \mathcal{E})$ is defined as follows.

$$d_\varphi(\widehat{\mathcal{G}}, \mathcal{G}) = \frac{\sum_{x \in \mathcal{S}, y \in \mathcal{V}\backslash\mathcal{S}} |e_{xy}|}{\min(vol(\mathcal{S}), vol(\mathcal{V}\backslash\mathcal{S}))}, \tag{11}$$

where $vol(\mathcal{S}) = \sum_{x \in \mathcal{S}, y \in \mathcal{V}} |e_{xy}|$ is the volume of subset $\mathcal{S}$ and $|e_{xy}|$ is the number of edges between $x$ and $y$.

In graph theory, the conductance of subset $\mathcal{S}$ is the ratio of edges going out to $\mathcal{V}\backslash\mathcal{S}$. This is related to the mixing time, which indicates how fast the probability distribution defined on subset $\mathcal{S}$ propagates to a nonzero probability for the entire node $\mathcal{V}$ along the Markov chain. If the conductance of the sampled subgraph is small, training subset $\mathcal{S}$ can be biased according to a subset of nodes that rarely propagates to external nodes. Although a small conductance enables to preserve the cluster information, it cannot help preserve the entire graph information. Consequently, conductance is not a suitable indicator for subgraphs used for learning instead of the original graph.

**(Graph Edit Distance)** Like the conductance $d_\varphi(\widehat{\mathcal{G}}, \mathcal{G})$, there is a measure to match two different graphs by counting the number of nodes and edges in traditional graph theory. Unlike the exact graph matching problem that solves graph isomorphism, the graph edit distance measures the similarity between two different graphs (*i.e.*, error-tolerant graph matching). Thus, the graph edit distance measures the minimum error of matching one graph to another.

$$d_{GED}(\widehat{\mathcal{G}}, \mathcal{G}) = \min_{(e_1, e_2, ..., e_k) \in \mathcal{P}} \sum_{i=1}^{k} c(e_i), \tag{12}$$

where $c(e_i)$ denotes the cost of each edit operation that includes vertex insertion, vertex deletion, edge insertion, and edge deletion. $\mathcal{P}(\widehat{\mathcal{G}}, \mathcal{G})$ denotes a set of edit paths. A single path consists of several edit operations $(e_1, e_2, ..., e_k)$ used to modify $\widehat{\mathcal{G}}$ to match another graph $\mathcal{G}$.

However, it is computationally expensive to find the optimal editing path. To find the optimal editing path at a low computation cost, we introduce the following assumptions. Suppose that graph $\mathcal{G} = (\mathcal{V}, \mathcal{E})$ is an unweighted graph. $\widehat{\mathcal{G}}$ is a subgraph that is defined for nodes $\mathcal{S} \subset \mathcal{V}$ and has only edges $\mathcal{E}_{\mathcal{S} \to \mathcal{S}} \subset \mathcal{S} \times \mathcal{S}$. All edit costs $c(e_i)$ are assumed to have values of $1$. Then, the graph edit distance between the subgraph $\widehat{\mathcal{G}}$ and the original graph $\mathcal{G}$ can be simplified as follows.

$$d_{GED}(\widehat{\mathcal{G}}, \mathcal{G}) = |\mathcal{V}\backslash\mathcal{S}| + |\mathcal{E}_{\mathcal{V}\backslash\mathcal{S} \to \mathcal{V}\backslash\mathcal{S}}| + |\mathcal{E}_{\mathcal{S} \to \mathcal{V}\backslash\mathcal{S}}|, \tag{13}$$

where $d_{GED}$ is calculated only using the number of nodes and edges in the subset. Although the conductance in (11) and graph edit distance in (13) contain structural information, they do not represent topological characteristics of the subgraphs.

**(Spectral Distance)** We typically compare graphs based on spectral analysis. There are several spectral methods [53], of which Laplacian is used to transform the domain of graph data. Let $A$ be the adjacency matrix that represents the graph $\mathcal{G}$, and $D$ be the degree matrix with diagonal elements defined by $D_{xx} = \sum_{y \in \mathcal{N}(x)} A_{xy}$. Then, the Laplacian matrix is defined as $L = D - A$, whereas the normalized Laplacian matrix is defined as $L = D^{-\frac{1}{2}} L D^{-\frac{1}{2}}$. Using the eigendecomposition of the Laplacian matrix, we can interpret the graph in the spectral domain instead of the spatial domain.

Suppose that the eigenvalues of Laplacian matrix $L$ is arranged in ascending order, *i.e.*, $0 = \lambda_1 \le \lambda_2 \le \cdots \le \lambda_{|\mathcal{V}|}$, with eigenvectors $\phi_1 \cdots \phi_{|\mathcal{V}|}$, where the eigenvalues $0 = \widehat{\lambda}_1 \le \widehat{\lambda}_2 \le \cdots \le \widehat{\lambda}_{|\mathcal{V}|}$ can be obtained from the Laplacian matrix $\widehat{L}$ of the subgraph $\widehat{\mathcal{G}}$ with common eigenvectors $\phi_1 \cdots \phi_{|\mathcal{V}|}$. Then, the spectral distance between the original graph $\mathcal{G}$ and the subgraph $\widehat{\mathcal{G}}$ is defined as follows.

$$d_\lambda(\widehat{\mathcal{G}}, \mathcal{G}) = \sqrt{\sum_{i=1}^{|\mathcal{V}|} (\lambda_i - \widehat{\lambda}_i)^2}. \tag{14}$$

Thus, if the adjacency matrix $A'$ of graph $\mathcal{G}'$ is defined by the permutation matrix $P$ (*i.e.*, $A' = P^T A P$), $\mathcal{G}'$ is isomorphic to $\mathcal{G}$ and $d_\lambda(\mathcal{G}', \mathcal{G}) = 0$. However, it is difficult to understand which nodes and edges should be sampled for good subgraph sampling by comparing the original and subgraphs through spectral distance. In contrast, we can intuitively interpret the proposed distributional metric in the context of good subgraph sampling and show that it is related to the curvature.

# B Curvature and Graph Neural Networks

**(Graph Diffusion Kernel)** Let $\mathcal{G} = (\mathcal{V}, \mathcal{E})$ be an undirected graph with nodes $\mathcal{V}$ and edges $\mathcal{E}$. The node features $X \in \mathbb{R}^{|\mathcal{V}| \times D}$ are defined for nodes $\mathcal{V}$ and the structure of $\mathcal{G}$ is represented as a matrix form $A$. The symmetric normalized graph Laplacian matrix can be defined as $L = D^{-\frac{1}{2}}(D - A)D^{-\frac{1}{2}}$. Then, the graph convolution is defined as $g_\psi(L) * x = U g_\psi(\Lambda) U^T x$ for spectral filtering $g_\psi(\Lambda) = \text{diag}(\psi)$, where $\psi \in \mathbb{R}^{|\mathcal{V}|}$ is a spectral coefficient and $x \in \mathbb{R}^{|\mathcal{V}| \times 1}$ is a graph signal. However, the spectral filtering is computationally expensive because it requires the eigendecomposition process $L = U \Lambda U^T$ to transform graph domain into signal domain. Thus, in [20, 9], approximated graph convolution has been presented using Chebyshev polynomials of Laplacian, *i.e.*, $U g_\theta(\Lambda) U^T x \approx (\sum_{k=0}^{K} \theta_k L^k)x$, where $\theta \in \mathbb{R}^{K+1}$ is a polynomial coefficient. Furthermore, many graph neural networks in [21, 41, 37, 22, 6, 35] are defined based on the diffusion matrix $\mathcal{T} = \sum_{k=0}^{\infty} \theta_k T^k$ with the diffusion coefficient $\theta$ and the transition matrix $T$. In diffusion-based methods, diverse kernels can be defined to aggregate and propagate the node features. For example, the first-order approximated graph convolution [20] uses the diffusion matrix $\mathcal{T} = \tilde{D}^{-\frac{1}{2}} \tilde{A} \tilde{D}^{-\frac{1}{2}}$, where $\tilde{A}$ is $I + A$, $\tilde{D}$ is a degree matrix of $\tilde{A}$. Random-walk, personalized page rank, and heat kernels use $(D^{-1}A)^k$, $\sum_{k=0}^{\infty} \alpha(1-\alpha)^k(D^{-1}A)^k$, and $\sum_{k=0}^{\infty} e^{-t}\frac{t^k}{k!}(D^{-1}A)^k$, respectively. In the case of the diffusion kernel based on non-symmetric normalized graph Laplacian $D^{-1}(D - A)$, it can be easily interpreted as the curvature.

Let $\mathcal{G} = (\mathcal{V}, w, m)$ be a graph, where $w$ and $m$ are edge weights and node measures, respectively. If $w(x,y) \in \{0,1\}$, $\mathcal{G}$ is referred to as the combinatorial graph with the corresponding adjacency matrix $A$. Then, the non-symmetric normalized graph Laplacian is discretized as follows. The Laplacian becomes a negative operator in the Riemannian manifold.

$$\Delta f(x) = \sum_{y \in \mathcal{V}} w(x,y)(f(y) - f(x)), \tag{15}$$

where $m(x) = \sum_{y \in \mathcal{N}(x)} w(x,y)$. Then, the probability measure [48] at each node can be defined by lazy random walk kernels.

$$m_x^\epsilon(y) := 1_y(x) + \epsilon \Delta 1_y(x), \tag{16}$$

where $m_x^\epsilon(y) = 1 - \epsilon$ if $y = x$, and $m_x^\epsilon(y) = \epsilon \cdot \frac{w(x,y)}{m(x)}$, otherwise. Then, the integral for the measure $m_x^\epsilon$ is defined as $\int f dm_x^\epsilon = \sum_{y \in \mathcal{V}} f(y) m_x^\epsilon(y) = (f + \epsilon \Delta f)(x)$. Based on this definition, the Wasserstein distance between $m_x^\epsilon, m_y^\epsilon$ can be defined in the following assumptions. Let $\mathcal{G} = (\mathcal{V}, w, m)$ be a graph and $x \neq y$ be nodes in the graph $\mathcal{G}$. In addition, $\nabla_{xy} f = \frac{f(x)-f(y)}{d(x,y)}$, and $\nabla_{xy} \Delta f = \frac{\Delta f(x) - \Delta f(y)}{d(x,y)}$. Then, the 1-Wasserstein satisfies the following equality for the 1-Lipschitz function $f$.

$$\mathcal{W}_1(m_x^\epsilon, m_y^\epsilon) = \sup_{f \in Lip(1)} \sum_{v \in \mathcal{V}} f(v)(m_x^\epsilon(v) - m_y^\epsilon(v)) = d(x,y) \sup_{f \in Lip(1)} \nabla_{yx}(f + \epsilon \Delta f). \tag{17}$$

The Ollivier-Ricci curvature $\kappa_{xy}$ is defined as the ratio of the distributional distance $\mathcal{W}_1(m_x^\epsilon, m_y^\epsilon)$ and the geodesic distance $d(x,y)$. In addition, the following generalized inequality holds for any 1-Lipschitz function $f$ without supremum:

$$\kappa_{xy}^\epsilon \leq 1 - \nabla_{yx}(f + \epsilon \Delta f). \tag{18}$$

Suppose that $\mathcal{G}$ is not a geometric graph, but a combinatorial graph with edges of length $d(x,y) = 1$ for all edges. Then, the matrix form is denoted as follows.

$$\kappa_{xy}^\epsilon \leq 1 - \nabla_{yx}\left[(I + \epsilon(D^{-1}A - I))X\right], \tag{19}$$

where $X$ represents $f$, and $I + \epsilon(D^{-1}A - I)$ represents first-order approximation of graph filter $\frac{1}{2}(I + D^{-1}A)$ when $\epsilon = \frac{1}{2}$.

**(Curvature graph networks)** If the GNNs are interpreted as diffusion-based graph neural networks in the aforementioned manner, the relationship between the curvature and the GNNs becomes clear. Thus, the curvature graph neural network [42] using the curvature as the kernel attention of the graph network has been proposed. The curvature measures how smoothly a message flows along the edge of the graph. Therefore, the negatively curved edges are likely to be inter edges of different communities. However, in [42], they design an indirect curvature attention network with mapping functions that can be learned by node features and node labels, because the curvatures only consider the structural information of the graph. Although the goal of the proposed method is to sample the subgraphs using only the structural information of graphs in large-scale graphs, we indirectly but experimentally show that the curvature contains useful information for learning graph networks.

# C Curvature and Substructures

**(Forman-Ricci curvature)** The curvature considers how flat an geometric object is. To measure the curvature, the metric space must be defined as $(\mathcal{X}, d)$, in which $\mathcal{X}$ is a underlying space and the distance function is $d : \mathcal{X} \times \mathcal{X} \to \mathbb{R}$. Fundamentally, the Riemannian manifold $(\mathcal{M}, g)$ for smooth manifolds $\mathcal{M}$ is equipped with the Riemannian metric $g$. The curvature $\kappa$ can be interpreted as one of the Riemannian metrics of the Riemannian manifold. In other words, the curvature can be considered as a

function that measures geometric quantities. Particularly, we focus on the Ricci-curvature, which can be defined in the discrete space (*e.g.*, graphs through the Ollivier-Ricci curvature [28] and Forman-Ricci curvature [10]). The detailed differences have been studied in [31].

The Ollivier-Ricci curvature is defined using the minimal transport cost between two points in a metric space. The Ricci-curvature can be discretized as a graph with the probability measure at each node. In contrast, the Forman-Ricci curvature is defined using topological arguments. It measures how fast the distance volume between the two points increases. Thus, it measures the dispersion rate of the geodesic. Then, the Forman-Ricci curvature is defined as follows.

$$\mathcal{F}(e_{xy}) = w_{e_{xy}} \left( \frac{w_x}{w_{e_{xy}}} + \frac{w_y}{w_{e_{xy}}} - \sum_{e_x \sim e_{xy}, e_y \sim e_{xy}} \left[ \frac{w_x}{\sqrt{w_{e_{xy}} w_{e_x}}} + \frac{w_y}{\sqrt{w_{e_{xy}} w_{e_y}}} \right] \right) \tag{20}$$

$$= w_{e_{xy}} \left( \frac{w_x}{w_{e_{xy}}} + \frac{w_y}{w_{e_{xy}}} - \left( \sum_{e_x \sim e_{xy}} \frac{w_x}{\sqrt{w_{e_{xy}} w_{e_x}}} + \sum_{e_y \sim e_{xy}} \frac{w_y}{\sqrt{w_{e_{xy}} w_{e_y}}} \right) \right), \tag{21}$$

where $w_{e_{xy}}$ is an weight of edge $e_{x,y}$ and $w_x, w_y$ are weights of nodes $x, y$. In addition, $e_x \sim e$ denotes the set of edges connected to $x$ except for $e_{xy}$ and $e_y \sim e$ denotes the set of edges connected to $y$ except $e_{xy}$.

The Forman-Ricci curvature has been used to find substructures [52] in the graph, because it is fast and scalable. However, despite these advantages, the association between structural errors of subgraphs and curvatures are unclear. Therefore, we attempt to explain the subgraph sampling via intuitive and descriptive Ollivier-Ricci curvatures. In this paper, we show that existing combinatorial subgraph sampling methods are closely related to the Ollivier-Ricci curvature.

**(Community)** Finding communities using the curvature has been studied in [26]. In this study, the communities was found through Ricci flow, which used the Ollivier-Ricci curvature to reduce the weight of positively curved edges and increase the weight of negatively curved edges. The increase in weight can be interpreted as an increase in the length of the edge; thus, the association between two nodes is reduced. Therefore, the communities can be found by removing the edges with the reduced association by the surgery in specific iterations.

$$w_{t+1}(e_{xy}) = (1 - \sigma) w_t(e_{xy}) - \sigma \cdot \kappa_t(e_{xy}) w_t(e_{xy}), \tag{22}$$

where $\sigma$ is an update weight of curvatures and $w_t$ denotes an weights at the $t$-step. Then, the edge weight $w_t(e_{xy})$ at time $t$ is updated to the edge weight $w_{t+1}(e_{xy})$ at time $t+1$ by the curvature $\kappa_t(e_{xy})$. As a result of the community separated by Ricci flow, we can show that positively (negatively) curved edges become intra-edges (inter-edges). Community detection problems have been widely studied for graph structural analysis and these characteristics have been used directly for subgraph sampling. For example, the cluster sampler obtains the samples of community-based subgraphs using a multi-level graph partitioning algorithm in the application of large-scale graphs. However, because the fixed number of partitions with the fixed number of nodes are sampled, it is difficult to obtain dynamic communities and find the optimal number of communities. Because neighbor samplers also sample a certain number of neighbors for each hop based on seed nodes, they can form the communities. However, because neighbors are randomly sampled, it is difficult to find hyper-parameters that construct suitable communities.

# D  Proofs

**(Proposition 1)** Let $w_x$ be a probability measure for node $x$ in a sampled subgraph, which can be defined as $w_x = \sum_{y \in \mathcal{V}} p(y|x) \delta_y$ by combinatorial decomposition, where $\sum_{y \in \mathcal{N}(x)} p(y|x) = 1, \delta_y = 1_y$. By Definition 2, the distributional distance $d_m(\mathcal{G}_x, \widehat{\mathcal{G}}_x)$ can be bounded by combinatorial decomposition as $d_m(\mathcal{G}_x, \widehat{\mathcal{G}}_x) \leq \sum_{y \in \mathcal{N}(x)} p(y|x) d_m(\mathcal{G}_x, v_y)$. Then, by the triangular inequality, the following inequalities hold.

$$d_m(\mathcal{G}, v_y) \leq d_m(\mathcal{G}_x, \mathcal{G}_y) + d_m(\mathcal{G}_y, v_y) = \mathcal{W}_1(m_x, m_y) + \mathcal{W}_1(m_y, \delta_y) \leq (1 - \kappa_{xy}) + 1 = 2 - \kappa_{xy}. \tag{23}$$

Therefore, the distributional distance between the original graph and sampled subgraph can be represented for node $x$ as follows.

$$d_m(\mathcal{G}_x, \widehat{\mathcal{G}}_x) \leq \sum_{y \in \mathcal{N}(x)} p(y|x)(2 - \kappa_{xy}) = 2 - \sum_{y \in \mathcal{N}(x)} p(y|x) \kappa_{xy}. \tag{24}$$

**(Corollary 1.1)** Let $\mathcal{G}_p = (\mathcal{V}, \mathcal{E})$ be a graph with positively curved edges $\kappa_{xy} \geq \kappa > 0$ for any edge $e_{xy}$ in $\mathcal{E}$. Then, the diffused probabilities of random-walk steps can be defined using curvature $\kappa$. Suppose that the probability distribution diffused $n$-hop through the random walk kernel reflects local structures of $n$-hop at each node. Then, the distributional distance between

the $n$-hop local structure $\mathcal{G}_x^{*n}$ with $n$-hop diffused probability measure $m_x^{*n}$ and the sampled local structure $\widehat{\mathcal{G}}_x$ with probability measure $w_x$ can be represented as follows.

$$d_m(\mathcal{G}_x^{*n}, \widehat{\mathcal{G}}_x) = \mathcal{W}_1(m_x^{*n}, w_x) \leq \sum_{y \in \mathcal{N}(x)} p(y|x) \left[ \mathcal{W}_1(m_x^{*n}, m_x^{*(n-1)}) + \cdots + \mathcal{W}_1(m_x, m_y) + \mathcal{W}_1(m_y, \delta_y) \right]. \quad (25)$$

As shown in [28], the following inequality holds, *i.e.*, $\mathcal{W}_1(\mu * m, \nu * m) \leq (1 - \kappa)\mathcal{W}_1(\mu, \nu)$. We use this inequality, $\mathcal{W}_1(m_x^{*(i+2)}, m_x^{*(i+1)}) \leq (1 - \kappa)\mathcal{W}_1(m_x^{*(i+1)}, m_x^{*i})$, to derive the followings.

$$d_m(\mathcal{G}_x^{*n}, \widehat{\mathcal{G}}_x) \leq \sum_{y \in \mathcal{N}(x)} p(y|x) \left[ (1-\kappa)^{n-1} + \cdots + (1-\kappa) + \mathcal{W}_1(m_x, m_y) + \mathcal{W}_1(m_y, \delta_y) \right] \quad (26)$$

$$= \sum_{y \in \mathcal{N}(x)} p(y|x) \left[ \frac{(1-\kappa)(1-(1-\kappa)^{n-1})}{1-(1-\kappa)} + \mathcal{W}_1(m_x, m_y) + 1 \right] \quad (27)$$

$$\approx \frac{(1-\kappa) - (1-\kappa)^n}{\kappa} + d_m(\mathcal{G}_x, \widehat{\mathcal{G}}_x). \quad (28)$$

**(Proposition 2)** In [28], $(\mathcal{V}, d)$ for $\epsilon$-geodesic satisfies that if $\kappa_{uv} \geq \kappa$ for any pair of nodes with $d(u,v) \leq \epsilon$, then $\kappa_{xy} \geq \kappa$ for any pair of nodes $x, y \in \mathcal{V}$. The Wasserstein distance can be represented using a duality form of $\mathcal{W}_1(\mu, \nu) = \sup_{f \in Lip(1)} \int_{\mathcal{V}} f d\mu - \int_{\mathcal{V}} f d\nu$. Then, the Wasserstein distance between two probability measures defined on two nodes is defined as follows.

$$\mathcal{W}_1(m_x, m_y) = \sup_{f \in Lip(1)} \sum_{v \in \mathcal{V}} f(v) \left( m_y(v) - m_x(v) \right) \quad (29)$$

$$= \sup_{f \in Lip(1)} \left[ (f(y) + \Delta f(y)) - (f(x) + \Delta f(x)) \right] \quad (30)$$

$$= d(x,y) \sup_{f \in Lip(1)} \nabla_{yx}(f + \Delta f). \quad (31)$$

The curvature of edge $e_{xy}$ is defined as $\kappa_{xy} = 1 - \frac{\mathcal{W}_1(m_x, m_y)}{d(x,y)} = \inf_{f \in Lip(1)}(1 - \nabla_{yx}(f + \Delta f))$. Let two geodesic paths $\mathcal{P}_i, \mathcal{P}_j$ be $x_i = v_0, v_1, \cdots, v_n = y_i$ for $\mathcal{P}_i$, $x_j = u_0, u_1, \cdots, u_n = y_j$ for $\mathcal{P}_j$. Because these two paths are the shortest paths between starting and ending nodes, the Wasserstein distance is computed as follows.

$$\mathcal{W}_1(m_{x_i}, m_{y_i}) \leq \sum_{k=0}^{n-1} \mathcal{W}_1(m_{v_k}, m_{v_{k+1}}) = \sum_{k=0}^{n-1} (1 - \kappa_{v_k, v_{k+1}}) d(v_k, v_{k+1}), \quad (32)$$

$$\mathcal{W}_1(m_{x_j}, m_{y_j}) \leq \sum_{k=0}^{n-1} \mathcal{W}_1(m_{u_k}, m_{u_{k+1}}) = \sum_{k=0}^{n-1} (1 - \kappa_{u_k, u_{k+1}}) d(u_k, u_{k+1}). \quad (33)$$

Therefore, it can be represented as follows.

$$\mathcal{W}_1(m_{x_i}, m_{y_i}) = \left( \inf_{f \in Lip(1)} \nabla_{y_i x_i}(f + \Delta f) \right) d(x_i, y_i), \quad (34)$$

$$\mathcal{W}_1(m_{x_j}, m_{y_j}) = \left( \inf_{f \in Lip(1)} \nabla_{y_j x_j}(f + \Delta f) \right) d(x_j, y_j). \quad (35)$$

Because two geodesic paths $\mathcal{P}_i, \mathcal{P}_j$ are the paths with length $d(x_i, y_i) = d(x_j, y_j) = n$ on common graphs, the following inequality holds for any 1-Lipschitz function $f$ that satisfies $\bar{\kappa} < 0$.

$$\nabla_{y_i x_i}(f + \Delta f) \times n - \nabla_{y_j x_j}(f + \Delta f) \times n = \sum_{k=0}^{n-1} (\kappa_{u_k u_{k+1}} - \kappa_{v_k v_{k+1}}) d(v_k, v_{k+1}) = \bar{\kappa} \times n, \quad (36)$$

where $\bar{\kappa} = \frac{1}{n} \sum_{k=0}^{n-1} (\kappa_{u_k u_{k+1}} - \kappa_{v_k v_{k+1}})$ is the mean of differences in curvatures of the path. Because the curvature $\kappa_{v_k v_{k+1}}$ of edges in $\mathcal{P}_i$ is larger than $\kappa_{u_k u_{k+1}}$ of edges in $\mathcal{P}_j$, the mean of difference $\bar{\kappa}$ is negative, *i.e.*, $\bar{\kappa} < 0$. Thus, the following inequality holds.

$$\nabla_{y_i x_i}(f + \Delta f) - \nabla_{y_j x_j}(f + \Delta f) < 0. \quad (37)$$

**(Table 1)** We present the approximated curvatures with 3-cycles in Definition 3 as follows.

$$\kappa_{xy} \geq -\left( 1 - \frac{1}{\mathsf{d}_x} - \frac{1}{\mathsf{d}_y} - \frac{\triangle_\sharp(x,y)}{\mathsf{d}_x \wedge \mathsf{d}_y} \right)_+ - \left( 1 - \frac{1}{\mathsf{d}_x} - \frac{1}{\mathsf{d}_y} - \frac{\triangle_\sharp(x,y)}{\mathsf{d}_x \vee \mathsf{d}_y} \right)_+ + \frac{\triangle_\sharp(x,y)}{\mathsf{d}_x \vee \mathsf{d}_y}, \quad (38)$$

where $\Delta_\sharp(x,y)$ denotes the number of triangles including the edge $e_{xy}$.

- **Edge sampler:** The edge sampler samples the subgraphs based on probabilities $p(e_{xy})$ defined on each edge $e_{xy}$. The probability model is obtained from the symmetric edge weight $w_{e_{xy}} = w_{e_{yx}}$, which is calculated by the sum of the degree normalized edge weights $w_{e_{xy}} \propto \frac{1}{\mathsf{d}_x} + \frac{1}{\mathsf{d}_y}$ for $e_{xy}$. This probability model can be interpreted as the probability proportional to the approximated curvature in the case of $\Delta_\sharp(x, y) = 0$. We assume that each node in the graph has a degree greater than 1 $\mathsf{d} > 1$. Then, (38) can be simplified as follows.

$$\kappa_{xy} \geq \widehat{\kappa}_{xy} = -\left(1 - \frac{1}{\mathsf{d}_x} - \frac{1}{\mathsf{d}_y}\right)_+ - \left(1 - \frac{1}{\mathsf{d}_x} - \frac{1}{\mathsf{d}_y} - \right)_+ = -2\left(1 - \frac{1}{\mathsf{d}_x} - \frac{1}{\mathsf{d}_y}\right). \tag{39}$$

Therefore, the edge sampler can be defined based on the weights proportional to the approximated Ollivier-Ricci curvature.

$$p(e_{xy}) \propto w_{e_{xy}} = \frac{\widehat{\kappa}_{xy} + 2}{2}. \tag{40}$$

- **Node sampler:** The node sampler also defines the probability model based on edge weights $w_{e_{xy}}$ like the edge sampler. Because the node sampler configures the subgraph by node-wise sampling, the probability $p(v_x)$ at each node $x$ is needed. Therefore, the probability model is determined in proportion to the node weights, which are the sum of connected edge weights as $p(v_x) \propto \left(\sum_{y \in \mathcal{N}(x)} \frac{1}{\mathsf{d}_y}\right)^2 = \left(\sum_{y \in \mathcal{N}(x)} w_{e_{xy}} - 1\right)^2 = \left(\sum_{y \in \mathcal{N}(x)}(\frac{1}{\mathsf{d}_x} + \frac{1}{\mathsf{d}_y}) - 1\right)^2$. Therefore, the node sampler also defines the probability model proportional to the approximated curvature without a cycle:

$$p(v_x) \propto \left(\sum_{y \in \mathcal{N}(x)} w_{e_{xy}} - 1\right)^2 = \left(\sum_{y \in \mathcal{N}(x)} \frac{\widehat{\kappa}_{xy} + 2}{2} - 1\right)^2. \tag{41}$$

- **Cluster sampler:** The cluster sampler is a multi-clusters sampler, which combines several clusters to configure a subgraph. As aforementioned, the curvature is related to intra-edges and inter-edges in the substructures in Section C as graph clusters. However, because the cluster sampler is for subset-unit sampling rather than for combinatorial sampling of minimum units such as nodes and edges, the probability model can be simplified through several assumptions for comparisons. The simplified probability model for the cluster sampler is clearly proportional to the approximated curvature.

In general, the local clustering coefficient (Watts-Strogatz) is defined as follows.

$$c(x) := \frac{|\text{ triangles in } \{\mathcal{N}(x) \cup x\}|}{|\text{ possible triangles in } \{\mathcal{N}(x) \cup x\}|} = \frac{1}{\mathsf{d}_x(\mathsf{d}_x - 1)} \sum_{y \in \mathcal{N}(x)} \Delta_\sharp(x, y). \tag{42}$$

The scalar curvature $\kappa_x$ is defined as $\kappa_x := \frac{1}{\mathsf{d}_x} \sum_{y \in \mathcal{N}(x)} \kappa_{xy}$. Then, in the case of d-regular graph, the scalar curvature can have the following lower bound [19]:

$$\kappa_x \geq \frac{1}{\mathsf{d}} \times \mathsf{d} \times \left(-2 + \frac{4}{\mathsf{d}} + \frac{3\Delta_\sharp(x, y)}{\mathsf{d}}\right). \tag{43}$$

The local clustering coefficient is also simplified as follows.

$$c(x) = \frac{1}{\mathsf{d}(\mathsf{d} - 1)} \times \mathsf{d} \times \Delta_\sharp(x, y). \tag{44}$$

Therefore, the number of triangles is represented as $\Delta_\sharp(x, y) = (\mathsf{d} - 1)c(x)$. Subsequently, the curvatures $\kappa_{xy}$ and clustering coefficient $c(x)$ can be associated:

$$\sum_{y \in \mathcal{N}(x)} (\kappa_{xy} + 2) \geq \sum_{y \in \mathcal{N}(x)} \left(\frac{4}{\mathsf{d}} + \frac{3(\mathsf{d} - 1)}{\mathsf{d}} c(x)\right) = 4 + 3(\mathsf{d} - 1)c(x). \tag{45}$$

Then, the node-wise probability model for the cluster sampler is defined as follows.

$$p(x) \propto c(x) \approx \frac{\sum_{x \in \mathcal{N}(x)}(\kappa_{xy} + 2) - 4}{3(\mathsf{d} - 1)}. \tag{46}$$

**(Proposition 3)** We represent the difference between the exact curvature and approximated curvature as the difference of the distributional distance. Suppose that the edge length is 1. Then, the difference can be defined as follow.

$$\|\kappa_{xy} - \widehat{\kappa}_{xy}\| = \widehat{\mathcal{W}}_1(m_x, m_y) - \mathcal{W}_1(m_x, m_y) \geq 0, \tag{47}$$

where $\widehat{\mathcal{W}}_1(m_x, m_y)$ denotes the approximated Wasserstein distance that is larger than the optimal distance $\mathcal{W}_1(m_x, m_y)$. We then present the upper bound $\widehat{\mathcal{W}}_1(m_x, m_y)$ of the distributional distance in the local structure $\{\mathcal{N}(x) \cup x\} \cup \{\mathcal{N}(y) \cup y\}$ as a

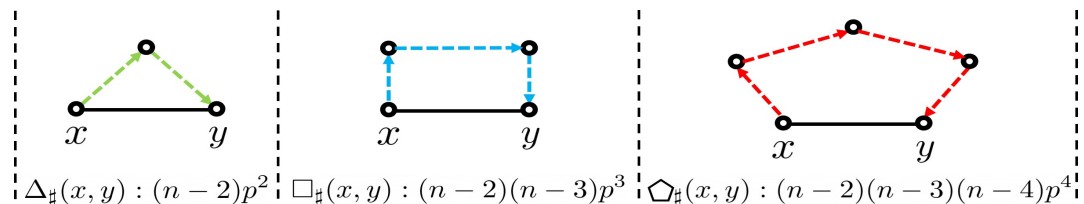

$$\Delta_\sharp(x,y) : (n-2)p^2 \quad \Box_\sharp(x,y) : (n-2)(n-3)p^3 \quad \bigcirc_\sharp(x,y) : (n-2)(n-3)(n-4)p^4$$

Figure 5: Let $\mathcal{G}(n,p)$ be the ER-graph (Erdős-Rényi model) with $n > 4$ nodes and edges connected by the probability $0 \leq p \leq 1$. Then, the number of cycles (3-cycles, 4-cycles, and 5-cycles) can be calculated probabilistically.

greedy calculated value. The distributional distance measures the distance transported from the neighboring nodes $u \in \mathcal{N}(x)$ of $x$ to the neighboring nodes $v \in \mathcal{N}(y)$ of $y$. Each transport cost is calculated by multiplying the transported distance $d(u,v)$ by the moved measure. If there is no path directly connected from each neighboring node of $x$ to a neighboring node of $y$ through a cycle, it moves through $x$ and $y$. Because the distributional distance is symmetric, we assume that $\mathsf{d}_y \geq \mathsf{d}_x > 1$ for convenience of the calculation.

If the cycle is not considered in the local structure around two nodes, the approximated distributional distance is calculated as follows.

$$\widehat{\mathcal{W}}_1(m_x, m_y) = 2 \times \frac{1}{\mathsf{d}_x} \times (\mathsf{d}_x - 1) + 0 \times \frac{1}{\mathsf{d}_x} - \frac{1}{\mathsf{d}_y} + 1 \times \frac{1}{\mathsf{d}_y} \times (\mathsf{d}_y - 1) = \left( 3 - \frac{2}{\mathsf{d}_x} - \frac{2}{\mathsf{d}_y} \right) = 1 - \widehat{\kappa}_{xy}. \tag{48}$$

It is the approximated distributional distance presented in (9). Then, if we calculate the approximated distributional distance considering the 3-cycles, we can compute $\widehat{\mathcal{W}}_1(m_x, m_y)$ more accurately:

$$\widehat{\mathcal{W}}_1(m_x, m_y) = 0 \times \frac{1}{\mathsf{d}_y} \times \Delta_\sharp + 1 \times \left( \frac{1}{\mathsf{d}_x} - \frac{1}{\mathsf{d}_y} \right) \times \Delta_\sharp + 2 \times \frac{1}{\mathsf{d}_x} \times (\mathsf{d}_x - \Delta_\sharp - 1) + 0 \times \frac{1}{\mathsf{d}_x} - \frac{1}{\mathsf{d}_y} + 1 \times \frac{1}{\mathsf{d}_y} \times (\mathsf{d}_y - \Delta_\sharp - 1) \tag{49}$$

$$= \left( 3 - \frac{2}{\mathsf{d}_x} - \frac{2}{\mathsf{d}_y} \right) - \left( \frac{\Delta_\sharp}{\mathsf{d}_x} + \frac{2\Delta_\sharp}{\mathsf{d}_y} \right) = 1 - \widehat{\kappa}_{xy}, \tag{50}$$

where $\Delta_\sharp$ denotes $\Delta_\sharp(x,y)$. It is also the approximated distributional distance in (3). As shown in this approximated distributional cost with 3-cycles, the distributional distance becomes more accurate as the number of cycles to be considered increases. The distributional distance in (49) is defined as the distance in (48) minus the distance shortened by the 3-cycles.

Then, we can consider how many cycles need to be considered to calculate the optimal transportation distance in the local structure $\{\mathcal{N}(x) \cup x\} \cup \{\mathcal{N}(y) \cup y\}$. If only the local structure is considered, the distance transported from the neighboring node of $x$ to the neighboring node of $y$ through the 6-cycle is the same as the distance transported through $x$ and $y$. Therefore, we can obtain the optimal transport distance limited to the local structure by considering until 5-cycles.

Subsequently, to calculate the distributional distances considering 4-cycles and 5-cycles, we have to consider all the cases where various cycles are included simultaneously. However, we simply consider the case of adding 4 and 5-cycles on no-cycle and the maximum distance that can be reduced. Then, maximally reduced distances by 4-cycles, 5-cycles are presented as follows.

$$\text{(maximal reduced distance by 4-cycles)} := -2 \times \frac{1}{\mathsf{d}_x} \times \Box_\sharp - 1 \times \frac{1}{\mathsf{d}_y} \times \Box_\sharp + 1 \times \frac{1}{\mathsf{d}_y} \times \Box_\sharp, \tag{51}$$

$$\text{(maximal reduced distance by 5-cycles)} := -2 \times \frac{1}{\mathsf{d}_x} \times \bigcirc_\sharp - 1 \times \frac{1}{\mathsf{d}_y} \times \bigcirc_\sharp + 2 \times \frac{1}{\mathsf{d}_y} \times \bigcirc_\sharp, \tag{52}$$

where $\Box_\sharp$ and $\bigcirc_\sharp$ abbreviate the number of 4-cycles $\Box_\sharp(x,y)$ and the number of 5-cycles $\bigcirc_\sharp(x,y)$, respectively. And let's assume that the degrees of nodes are also the same $\mathsf{d}_x = \mathsf{d}_y = (n-1)p$. As a result, the error between the optimal distributional distance and the approximated distributional distance with 3-cycles is presented as the upper bound.

$$\|\kappa_{xy} - \widehat{\kappa}_{xy}\| \leq \text{(maximal reduced distance by 4-cycles)} + \text{(maximal reduced distance by 5-cycles)} \tag{53}$$

$$= \frac{1}{\mathsf{d}}(2\Box_\sharp + \Box_\sharp - \Box_\sharp + 2\bigcirc_\sharp + \bigcirc_\sharp - 2\bigcirc_\sharp) = \frac{2}{\mathsf{d}}\Box_\sharp + \frac{1}{\mathsf{d}}\bigcirc_\sharp \tag{54}$$

$$= \frac{2}{(n-1)p}(n-2)(n-3)p^3 + \frac{2}{(n-1)p}(n-2)(n-3)(n-4)p^4 \tag{55}$$

$$\leq \frac{2\mathsf{d}^2 p + 2\mathsf{d}^3 p}{\mathsf{d}} = 2\mathsf{d}p + 2\mathsf{d}^2 p. \tag{56}$$

# E    Algorithm Details

**(Graph Coarsening-based Sampling)** Graph coarsening is to sample the edges to make the graph sparse. In this sampling method, the number of nodes is not reducible but only edges can be removed until the coarsened graph is out of bound in the defined error. Even if independent nodes without any edge are considered to remove, the method hardly samples the subgraphs with a similar number of nodes. This method can be used to merge linked nodes into a hyper node to reduce the size of the subgraph. However, it is not proper for node classification tasks, which needs to classify each node.

**(Graph Partitioning-based Sampling)** Graph partitioning is to split the original graph into parts. Because minimizing losses at partition boundaries has priority, there is an association between graph partitioning and clustering. In the sense of clustering, the main objectives of this problem is the minimization of outer edges and maximization of inner edges. Therefore, a good partitioned graph tends to be clustered to have localized subgraphs, which makes each subgraph to be biased toward a specific neighborhood. Although it depends on the characteristics of the graph, learning with biased subgraphs is likely to have the same side effects as learning with biased mini-batches.

**(Graph Covering-based Sampling)** Graph covering consists of vertex and edge covers, in which edges and nodes are covered by the vertex and edge covers, respectively. If the vertex cover is sampled to cover as many edges as possible, this vertex cover can induce a subgraph with a small number of nodes. However, finding the minimum vertex cover is an NP-hard optimization problem. The nodes of the vertex cover are not suitable for the subgraph sampling method, because they are very sparse or independent.

**(Graph Combinatorial Sampling)** Graph combinatorial sampling is to configure the graphs by combining elements such as nodes and edges. Because each element is sampled independently according to the probability model, it is difficult to use global structural information compared with other methods. Therefore, it is not possible to sample the subgraphs that are optimal for a particular purpose. However, sampling can be done very efficiently even on large-scale graphs.

The proposed method presents a probability model proportional to the curvature that can reflect local structural information to reduce the distributional distance from the original graph. Because our method has the form of combinatorial sampling, the optimal substructure for the local structure can be sampled. However, it is impossible to sample globally optimal substructures. Therefore, we set up an initial seed node, in which the locally approximated structure is distributed throughout the original graph. The randomly sampled nodes $x \sim \mathbf{U}$ with uniform probabilities can be used as the initial seed node. Alternatively, nodes with a large scalar curvature $x \propto \frac{1}{d_x} \sum_{y \in \mathcal{N}(x)} \kappa_{xy}$ can be set as seed nodes and nodes with a large sum of degree-normalized curvatures can be set as seed nodes. We observed that the seed nodes sampled based on the scalar-curvature make the sampled subgraphs with large curvatures and induce good performance on certain datasets. However, the stable performance is obtained when setting nodes with a large sum of degree-normalized curvatures as seed nodes. Thus, in our method, seed nodes are obtained through the following probability model $p(x|\mathcal{V}) \propto \sum_{y \in \mathcal{N}(x), y \in \mathcal{V}} p(x|y)$, where $p(x|y)$ is the degree-normalized curvature $\frac{1}{d_y} \kappa_{xy}$. We assume that the seed nodes $\mathcal{S}_0$ are distributed over the entire structure. Then, we recalculate the curvature-based probability model $p(x|\mathcal{S}_0) \propto \sum_{y \in \mathcal{N}(x), y \in \mathcal{S}_0} p(x|y)$ for approximating the local structure $\mathcal{N}(\mathcal{S}_0)$ around the seed node $\mathcal{S}_0$. This step-wise sampling is performed to minimize the combinatorial distributional distance in Definition 2 through the conditional probability model $p(x|\mathcal{S}_0)$. The proposed method sets the number of steps $t$ as a hyperparameter. Increasing the number of steps reduces the number of seed nodes $|\mathcal{S}_0|$. By refraining from being widely distributed across the entire structure, more accurate approximated local structures can be obtained. Newly sampled combinatorial components $\{u_0, u_1, \cdots, u_{[m/s]}\}$ of approximating the local structure $\mathcal{N}(\mathcal{S}_i)$ at each step are included in structural components of the subgraph $\mathcal{S}_{i+1} = \mathcal{S}_i \cup \{u_0, u_1, \cdots, u_{[m/s]}\}$. The substructure including all the nodes is sampled, which becomes the subgraph at the last step.

# F    Additional Experiments

**(Environment)** The experiments were conducted on a machine equipped with a Intel Core i9-10980XE CPU @ 3.00GHz, NVIDIA GeForce RTX 3090, and 256GB DDR4 memory. We used Pytorch 1.9.0 with CUDA 11.1 and CUDNN 8.0.5.

**(Sampler details)** We compared nine samplers including the proposed method. Among them, random walk sampler, cluster sampler, ppr sampler, neighbors sampler, and the proposed sampler can tune hyperparameters.

- Random walk - walks : {2,4,6,8,10}
- Cluster - parts : {10000,20000,40000,80000,100000}
- Personalized Page Rank - topk
- Neighbor - hop, neighbors
- LoCur - steps : {1,2,3,4,5,6}

695  For all experiments, the optimal hyperparameters for each dataset was determined empirically.

696  Table 5 shows the time for sampling one graph using each sampler for all datasets. In Table 6, preprocessing means that it
697  is performed on the entire step in an initial step, like partitioning for cluster sampler and computation of curvature for the
698  proposed method. However, once these are preprocessed, sampling does not occur in the middle of training. Thus, it is negligible
699  compared to total training time. When training for semi-supervised node classification tasks with the ogbn-arxiv graph 1%
700  setting, memory can be saved by 36% compared to the case of using the entire graph, while the performance is reduced by only
701  0.03% if the proposed sampler is used.

Table 5: Sampling time (sec).

| Task | Dataset | sampling ratio (%) | random | neighbor | node | edge | random walk | cluster | ppr | LoCur (ours) |
|---|---|---|---|---|---|---|---|---|---|---|
| | ogbn-arxiv | 1 | 0.00713 | 0.21635 | 0.00811 | 0.04110 | 0.00635 | 0.01503 | 1.01422 | 0.03259 |
| | ogbn-arxiv | 5 | 0.00784 | 0.23973 | 0.01245 | 0.04326 | 0.01139 | 0.04891 | 1.04189 | 0.03986 |
| Node classification | ogbn-arxiv | 10 | 0.00932 | 0.25723 | 0.01660 | 0.04750 | 0.01528 | 0.09224 | 1.04576 | 0.04676 |
| | ogbn-mag | 1 | 0.03867 | 1.05332 | 0.04227 | 0.21121 | 0.03882 | 0.07359 | 6.80543 | 0.23211 |
| | ogbn-mag | 5 | 0.04313 | 1.17195 | 0.06083 | 0.23049 | 0.04992 | 0.20697 | 6.77334 | 0.26773 |
| | ogbn-mag | 10 | 0.04936 | 1.29256 | 0.08332 | 0.25149 | 0.06205 | 0.45105 | 6.78215 | 0.35181 |
| | DD | 20 | 0.00068 | 0.00358 | 0.00132 | 0.00129 | 0.00091 | 0.00474 | 0.14471 | 0.00381 |
| Graph classification | REDDIT-B | 20 | 0.00040 | 0.00176 | 0.00085 | 0.00055 | 0.00060 | 0.00590 | 0.15041 | 0.00160 |
| | REDDIT-5K | 20 | 0.00043 | 0.00147 | 0.00060 | 0.00052 | 0.00056 | 0.00568 | 0.14438 | 0.00184 |
| | COLLAB | 50 | 0.00035 | 0.00146 | 0.00049 | 0.00046 | 0.00051 | 0.00570 | 0.14296 | 0.00162 |

Table 6: Preprocessing time (sec).

| Task | Dataset | #nodes | #edges | random | neighbor | node | edge | random walk | cluster | ppr | LoCur (ours) |
|---|---|---|---|---|---|---|---|---|---|---|---|
| Node classification | ogbn-arxiv | 169,343 | 1,166,243 | 0.01238 | 0.01581 | 0.06482 | 0.05727 | 0.15548 | 3.04642 | 0.01228 | 30.40937 |
| | ogbn-mag | 1,939,743 | 21,111,007 | 0.06353 | 0.11086 | 0.42007 | 0.36620 | 0.84027 | 16.71598 | 0.06209 | 31.07691 |
| | DD | 284 | 694 | 0.00010 | 0.00025 | 0.00088 | 0.00036 | 0.00094 | 0.00024 | 0.00010 | 0.00314 |
| Graph classification | REDDIT-B | 400 | 455 | 0.00010 | 0.00021 | 0.00104 | 0.00031 | 0.00057 | 0.00019 | 0.00010 | 0.00307 |
| | REDDIT-5K | 508 | 618 | 0.00010 | 0.00023 | 0.00130 | 0.00036 | 0.00058 | 0.00030 | 0.00010 | 0.00259 |
| | COLLAB | 75 | 1179 | 0.00010 | 0.00021 | 0.00069 | 0.00030 | 0.00061 | 0.00020 | 0.00010 | 0.00307 |

702  **(Labeling Node Classification)** We present a new labeling node classification task, because existing node classification tasks
703  are not suitable for subgraphs sampled by samplers. In existing semi-supervised node classification setting, train, valid, and test
704  nodes were pre-split for the entire graph. Therefore, although good subgraphs are sampled, their performance is determined by
705  the number of train nodes, which is pre-defined for the entire graph, in the subgraph. To avoid this problem, we propose a new
706  task that samples only one subgraph, uses the subgraph for training, and tests all nodes of the entire graph. In this way, the
707  generalization performance of the sampled subgraph can be evaluated without external factors.

708  In this experiment, we evaluated several subgraph samplers with four GNN models using three datasets. For training, we used
709  the Adam optimizer and the mean of the results obtained through ten runs was used as a performance index. The random seed
710  was set to 1000 to enable reproducibility. For node classification tasks, gradient was updated per every iteration (per subgraph).
711  The experimental setting for each GNN is the same across the datasets. If we used full samplers, the epoch was set to 500. If
712  not, the epoch was set to 200.

713  The dataset descriptions for node classification tasks are given in Table 7. The details of the GNN models can be found in Table
714  8. Please note that when training GAT for the ogbn-mag dataset, the number of attention head was set to 1 due to the memory
715  issue.

Table 7: Node classification datasets.

| Dataset | #Nodes | #Edges | #Classes | Node.dim |
|---|---|---|---|---|
| ogbn-arxiv | 169,343 | 1,166,243 | 40 | 128 |
| ogbn-mag | 1,939,743 | 21,111,007 | 349 | 128 |

Table 8: Node classification baseline configurations.

| Baseline | Training | | | Model config | | |
|---|---|---|---|---|---|---|
| | lr | dropout | #epoch | hidden dim | #layers | #attention heads |
| GCN | 0.01 | 0.5 | 200 | 256 | 3 | - |
| GraphSAGE | 0.01 | 0.5 | 200 | 256 | 5 | - |
| GCNII | 0.001 | 0.1 | 200 | 256 | 18 | - |
| GAT | 0.01 | 0.5 | 200 | 128 | 3 | 4 |

(**Semi-supervised Node Classification**) For this experiment, we followed general semi-supervised node classification settings. However, because of the aforementioned problem, we sampled 100 subgraphs when sampling 1% of nodes. The datasets and GNN settings are the same as those of the labeling task. We present numerical results in Table 9.

Table 9: Semi-supervised node classification.

| node | ogbn-arxiv/GAT | | ogbn-arxiv/GCNII | | ogbn-mag/GCNII | | ogbn-mag/GAT | |
| classification | Valid acc (%) | Test acc (%) | Valid acc (%) | Test acc (%) | Valid acc (%) | Test acc (%) | Valid acc (%) | Test acc (%) |
| --- | --- | --- | --- | --- | --- | --- | --- | --- |
| random | 63.50±0.9051 | 62.92±1.5138 | 67.94±0.3953 | 67.28±0.8320 | 31.61±0.4972 | 32.17±0.6322 | 27.65±0.4529 | 29.11±0.5694 |
| neighbor | **68.76±0.4069** | **68.20±0.5315** | 70.43±0.2307 | 69.77±0.5997 | 34.04±0.6178 | 34.13±0.8366 | 30.66±0.6429 | 31.68±0.9938 |
| node | 67.20±0.5178 | 66.81±0.6102 | 66.39±0.3706 | 65.71±0.6186 | 30.05±0.4664 | 30.99±0.5589 | 26.27±0.8824 | 27.65±1.1153 |
| edge | 67.31±0.5181 | 66.84±0.7397 | 69.66±0.2105 | 68.95±0.5871 | 31.97±0.4184 | 32.21±0.5373 | 30.40±0.7125 | 31.63±0.7575 |
| random walk | 67.50±0.3482 | 67.38±0.6506 | 70.18±0.3671 | 69.46±0.6359 | 32.75±0.5707 | 32.80±0.6941 | 30.76±0.7666 | 31.91±0.9151 |
| cluster | 66.12±0.8913 | 65.76±1.1373 | **71.55±0.2047** | **70.77±0.6731** | **35.17±0.5612** | **35.46±0.7138** | 31.78±0.7862 | 32.91±0.7804 |
| ppr | 66.06±0.4956 | 65.02±0.7743 | 69.21±0.3486 | 67.68±0.9033 | 32.23±0.4246 | 32.50±0.4054 | 28.13±0.5190 | 29.50±0.5450 |
| **LoCur (Ours)** | 68.71±0.4200 | 67.74±0.7181 | 71.05±0.3059 | 70.43±0.4124 | 34.92±0.7565 | 34.90±1.1214 | **31.96±1.0061** | **33.11±1.1926** |
| original graph | 72.11±0.0679 | 71.12±0.2742 | 73.78±0.0934 | 72.55±0.2465 | N/A | N/A | N/A | N/A |

(**Graph classification**)

Unlike node classification tasks, graph classification typically uses the graphs without preprocessing, because the number of nodes is not as large as the graph dataset for node classification. However, even in this case, advantages of using sampled subgraph for graph classification are clear.

First, because the number of nodes in the input graphs is limited to the sample size, training can be facilitated by preventing the fluctuation of memory usage. The size of the graphs constituting the dataset for graph classification is not consistent. For example, the average number of nodes of the graphs in the DD dataset is 284, while the number of nodes in the largest graph is 5748. The variable size of input causes fluctuations in memory usage, especially if the device is constrained.

Second, training time can be considerably reduced. The large dimension of the node feature induces the large training time saving. In addition, it is more evident when we use social network datasets. For the social network datasets,the node degree is typically used as the node feature in the form of one hot vector, because there is no given node feature. Thus, the dimension of the node feature is the largest degree among the graphs of the whole dataset. However, if we use the subgraphs with a limited number of nodes as training data, the maximum of degree cannot exceed the sample size. In this case, by reducing the dimension of the node feature, we can reduce the computational cost. For instance, the REDDIT-BINARY dataset has a max degree of 3782, and the degree is used as the node feature in the one-hot vector form. Instead of using the 3782-dimensional node features, we can use only 80-dimensional node features, if we set the sample size of the subgraph to 80.

Third, although a new graph with arbitrary size is given, the trained GNN can work well. As aforementioned, when using the node degree as a node feature for the social datasets, the dimension of the node feature is set to the max degree across the whole training graphs. However, if a new graph with a degree that is greater than the max degree is given, the inference is impossible. In contrast, if the feature dimension is bound to the sample size, no problem occurs.

Graph classification tasks predict the label assigned to the entire graph by grasping the whole structure of the graph. Therefore, it is very different from node classification tasks, where the local context near the nodes is important to predict the label assigned to each node. In general, the graph classification task includes the pooling stage, in which each node feature is aggregated to create a graph-level feature, and commonly used methods are mean pooling and sum pooling. Due to the existence of the pooling stage, when sampling the subgraphs for graph classification, we need to sample various nodes that are important either in the global context or in the local context.

To grasp the graph structure for graph classification, many studies [50, 55, 51] have been reported, which consider the entire graph as a set of building blocks such as subgraph or motif. The Mesoscopic structure can be captured by finding a subgraph or motif that preserves local properties. In [55], global structures were represented by considering the interaction between these local structures. Therefore, the following conclusion can be drawn. In subgraph sampling for graph classification, which includes a special process called pooling, it is necessary to evenly sample intra-motif nodes, which captures local structures as well as inter-motif nodes that play an important role in the connection between motifs.

It is challenging to sample the subgraphs so as to preserve the global context while maintaining the local structure of the original graph with existing sampling methods. However, because our sampling probability model is based on the curvature, we can sample a subgraph that satisfies both conditions.

In [49, 26], it was examined how the curvature in the graph represents the local and global characteristics of the graph. In particular, in [49], it was described how ricci curvature was related to global centrality and local properties, respectively. If the curvature is negative and smaller, edges connect motifs for global connectivity. Conversely, if the curvature is positive and the clustering coefficient is high, edges exist inside the motifs. Therefore, to sample the subgraphs that can preserve the global and local structure, the negatively and positively curved edges should be evenly selected. However, because the sampling size is limited, we select the most negative and positive edges to form a subgraph. Therefore, the proposed method is designed to sample in proportion to the square of the deviation from the average curvature.

The benchmark datasets used for graph classification can be divided into two types, bioinformatics dataset and social datasets. Among them, we selected datasets whose average number of nodes were relatively large to demonstrate the effectiveness of the proposed subgraph sampling method. Therefore, REDDIT-BINARY, REDDIT-MULTI-5K, and COLLAB[56] were used for the social datasets, and DD[46] was used for the bioinformatics dataset. For the DD,REDDIT-BINARY and REDDIT-5K datasets, the subgraph sample size was set to 1/5 of the average number of nodes in the dataset. Because the graphs in the COLLAB dataset are small, the sampling ratio was set to 1/2. Table 10 describes the datasets used for this experiment.

Table 10: Graph classification datasets.

| Dataset | #Graphs | #Classes | Avg.Nodes | Avg.Edges | Max.node | Node.dim | Sample size |
|---------|---------|----------|-----------|-----------|----------|----------|-------------|
| DD | 1,178 | 2 | 284.32 | 715.66 | 5748 | 89 | 50 |
| REDDIT-B | 2,000 | 2 | 429.63 | 497.75 | 3782 | - | 80 |
| REDDIT-5K | 4,999 | 5 | 508.52 | 594.87 | 3648 | - | 100 |
| COLLAB | 5,000 | 3 | 74.49 | 2457.78 | 492 | - | 30 |

The performance was evaluated using 10-fold cross validation according to [54, 47]. In the DD dataset, which is a bioinformatic dataset, node features were given. For graph-level pooling, SUM pooling was used as a readout function. Batch size and epoch were set to 32 and 200, respectively. For three social datasets, we used the node degree as features in the form of one-hot vectors according to [54, 47], because there were no node features. For graph-level pooling, MEAN pooling was used as a readout function, and batch size and epoch were set to 128 and 350, respectively. For the cluster sampler, in which 'parts' should be set as a hyperparameter (*i.e.*, how many parts to view the entire graph). We set the 'parts' to be determined by the size of the graph (number of nodes) regardless of the dataset. Table 11 contains the details of several baselines for graph classification.

Table 11: Configurations of graph classification baselines.

| Type | Dataset | Baseline | Training | | | | Model config | | |
|------|---------|----------|----|---------|--------|------------|------------|---------|------------------|
| | | | lr | dropout | #epoch | batch size | hidden dim | #layers | #attention heads |
| Bioinformatics | DD | GCN | 0.01 | 0.5 | 200 | 32 | 32 | 2 | - |
| | | GraphSAGE | 0.01 | 0.5 | | | | 4 | - |
| | | GCNII | 0.001 | 0.5 | | | | 17 | - |
| | | GIN-0 | 0.01 | 0.5 | | | | 4 | - |
| | | GAT-GC | 0.01 | 0 | | | | 4 | 1 |
| Social network | REDDIT-B REDDIT-5K COLLAB | GCN | 0.01 | 0.5 | 350 | 128 | 64 | 2 | - |
| | | GraphSAGE | 0.01 | 0.5 | | | | 4 | - |
| | | GCNII | 0.001 | 0.5 | | | | 17 | - |
| | | GIN-0 | 0.01 | 0.5 | | | | 4 | - |
| | | GAT-GC | 0.01 | 0 | | | | 4 | 1 |

Table 12 shows graph classification accuracy according to the sampling ratio. As shown in the table, the proposed method exhibits the state-of-the-art performance, regardless of the sampling ratio. Tables 13, 14, 15, and 16 show graph classification accuracy on the DD, REDDIT-BINAR, REDDIT-MULTI-5, and COLLAB datasets, respectively. These comparisons consistently show the effectiveness of the proposed method.

Table 12: Graph classification accuracy according to sampling ratio.

| GCN | DD/10% | DD/20% | REDDIT-B/10% | REDDIT-B/20% | REDDIT-5K/10% | REDDIT-5K/20% | COLLAB/20% | COLLAB/50% |
|-----|--------|--------|--------------|--------------|---------------|---------------|------------|------------|
| random | 76.74±3.5290 | 79.44±4.6153 | 78.07±3.1554 | 81.38±2.8620 | 34.64±1.4852 | 38.41±1.6066 | 64.98±1.3776 | 69.34±1.6663 |
| neighbor | 69.75±3.2693 | 72.42±2.1710 | 78.45±2.3932 | 82.99±2.2842 | 41.39±1.6787 | 46.99±1.7218 | 64.94±2.0262 | 68.77±1.9170 |
| node | **77.66±3.3531** | 78.95±4.6996 | **87.71±2.1584** | 89.34±1.6096 | 45.94±1.5994 | **50.34±1.5081** | 64.85±1.9100 | 69.01±2.0218 |
| edge | 75.25±2.4826 | 74.45±2.1898 | 81.13±1.7302 | 87.08±2.1656 | 39.39±2.0594 | 46.56±1.4944 | 64.38±1.9147 | 69.03±1.7245 |
| random walk | 69.08±2.2111 | 72.83±3.5998 | 80.17±1.5837 | 84.37±1.5300 | 40.46±1.4986 | 45.95±1.6980 | 64.90±1.2438 | 69.25±1.2352 |
| cluster | 66.07±2.1793 | 66.00±3.4210 | 69.41±3.3951 | 72.86±2.3574 | 31.71±2.1585 | 33.84±1.1456 | 56.55±0.5435 | 62.11±1.8089 |
| ppr | 76.46±3.4249 | 75.15±4.2903 | 79.68±2.7840 | 80.99±2.2588 | 42.60±1.8427 | 46.10±1.3539 | 60.42±1.5170 | 66.52±1.7044 |
| **LoCur (Ours)** | 77.17±2.8647 | **79.56±4.3048** | 87.55±1.5493 | **90.12±1.1265** | **46.08±2.3627** | 49.99±1.5247 | **65.40±1.7117** | **69.48±1.7263** |

Table 13: Graph classification accuracy on DD (%).

| DD | GCN | GraphSAGE | GCNII | GIN-0 | GAT-GC |
|---|---|---|---|---|---|
| random | 79.44±4.6153 | **79.00±4.3275** | **78.05±2.8671** | 76.88±4.8735 | **78.39±5.0960** |
| neighbor | 72.42±2.1710 | 73.01±4.2728 | 73.22±3.5919 | 71.75±2.0162 | 75.36±3.3472 |
| node | 78.95±4.6996 | 77.89±4.2335 | 77.56±3.8740 | 76.01±4.2048 | 78.07±4.2521 |
| edge | 74.45±2.1898 | 73.44±3.2937 | 72.04±2.2719 | 72.00±2.6178 | 75.12±1.5757 |
| random walk | 72.83±3.5998 | 71.31±2.8796 | 72.81±2.6505 | 72.58±3.5552 | 73.40±3.6231 |
| cluster | 66.00±3.4210 | 64.64±2.4281 | 65.35±2.1771 | 66.22±2.8087 | 65.10±2.6852 |
| ppr | 75.15±4.2903 | 75.56±4.6815 | 74.13±4.1562 | 74.55±3.7862 | 75.97±4.1399 |
| **LoCur (Ours)** | **79.56±4.3048** | 78.38±4.0667 | 77.02±2.9737 | **77.32±4.0964** | 78.26±3.3836 |
| original graph | 81.35±2.9971 | 82.88±4.0471 | 77.99±4.1586 | 80.16±3.4829 | 81.29±3.3207 |

Table 14: Graph classification accuracy on REDDIT-BINARY (%).

| REDDIT-B | GCN | GraphSAGE | GCNII | GIN-0 | GAT-GC |
|---|---|---|---|---|---|
| random | 81.38±2.8620 | 81.78±2.9289 | 80.45±3.5311 | 81.40±2.6747 | 82.15±2.7213 |
| neighbor | 82.99±2.2842 | 83.96±1.8825 | 84.56±2.2474 | 84.48±1.4364 | 84.96±1.8539 |
| node | 89.34±1.6096 | 90.15±1.3836 | 89.33±1.6985 | 89.73±1.6502 | 90.28±1.4688 |
| edge | 87.08±2.1656 | 87.20±1.6394 | 85.99±1.6241 | 86.64±1.6287 | 87.23±2.3865 |
| random walk | 84.37±1.5300 | 85.30±1.1102 | 81.85±1.8239 | 83.83±1.8106 | 84.82±2.1063 |
| cluster | 72.86±2.3574 | 71.91±2.3884 | 71.84±3.3214 | 74.09±2.1185 | 73.89±2.8110 |
| ppr | 80.99±2.2588 | 81.73±2.2829 | 80.71±1.9299 | 83.07±1.5707 | 84.90±2.3119 |
| **LoCur (Ours)** | **90.12±1.1265** | **90.94±1.2293** | **90.14±0.6997** | **90.24±1.0670** | **91.19±1.2035** |
| original graph | 81.61±2.1039 | 79.81±2.4883 | 84.73±2.3256 | 86.91±1.9735 | 92.55±2.1038 |

Table 15: Graph classification accuracy on REDDIT-MULTI-5K (%).

| REDDIT-5K | GCN | GraphSAGE | GCNII | GIN-0 | GAT-GC |
|---|---|---|---|---|---|
| random | 38.41±1.6066 | 38.23±1.4061 | 38.03±1.8110 | 37.95±1.5313 | 38.36±1.3094 |
| neighbor | 46.99±1.7218 | 47.40±1.2698 | 46.39±0.8856 | 46.25±0.4337 | 46.69±1.0210 |
| node | **50.34±1.5081** | 50.15±1.9532 | **50.11±1.7265** | **50.54±1.5885** | **50.30±1.5571** |
| edge | 46.56±1.4944 | 46.89±1.2716 | 45.28±1.4890 | 46.11±1.6029 | 46.03±2.1224 |
| random walk | 45.95±1.6980 | 46.63±1.6052 | 45.49±1.5282 | 45.65±1.4225 | 45.60±1.7709 |
| cluster | 33.84±1.1456 | 34.64±1.8862 | 32.69±1.6914 | 33.89±1.2318 | 35.12±1.4595 |
| ppr | 46.10±1.3539 | 46.64±1.5032 | 46.70±1.1571 | 46.26±1.6166 | 47.09±1.3578 |
| **LoCur (ours)** | 49.99±1.5247 | **50.36±1.5384** | 50.07±1.4427 | 49.71±1.5194 | 50.16±1.8804 |
| original graph | 49.04±1.5736 | 48.45±1.1268 | 50.36±1.4778 | 50.38±1.2109 | 58.17±4.6943 |

Table 16: Graph classification accuracy on COLLAB (%).

| COLLAB | GCN | GraphSAGE | GCNII | GIN-0 | GAT-GC |
|---|---|---|---|---|---|
| random | 69.34±1.6663 | 69.28±1.3070 | 69.30±1.5405 | 68.95±1.3834 | 68.12±1.4702 |
| neighbor | 68.77±1.9170 | 68.56±1.5034 | 68.53±0.9320 | 67.87±1.1613 | 67.47±1.2766 |
| node | 69.01±2.0218 | 69.70±1.8717 | 69.29±1.7179 | 69.12±1.8614 | 68.42±1.9550 |
| edge | 69.03±1.7245 | 68.91±1.5967 | 69.30±1.4742 | 68.20±1.8398 | 68.41±1.4364 |
| random walk | 69.25±1.2352 | 68.97±1.2933 | 68.31±1.4007 | 68.91±1.0469 | 68.54±1.0148 |
| cluster | 62.11±1.8089 | 61.47±1.4807 | 61.16±1.5676 | 61.85±1.2656 | 61.19±1.3943 |
| ppr | 66.52±1.7044 | 66.98±1.4292 | 66.22±1.8055 | 65.76±1.3993 | 66.04±1.1868 |
| **LoCur (Ours)** | **69.48±1.7263** | **70.04±1.8101** | **69.61±1.7005** | **69.33±1.6915** | **69.20±1.2228** |
| original graph | 84.22±1.4804 | 83.57±1.7301 | 84.33±1.6665 | 84.35±1.3020 | 91.91±3.7769 |

778 **(Visualization)** Figs.6 and 7 visualize the subgraphs produced by several different samplers. As shown in the figures, the
779 proposed method samples the subgraphs to include representative structures without bias compared to other methods.

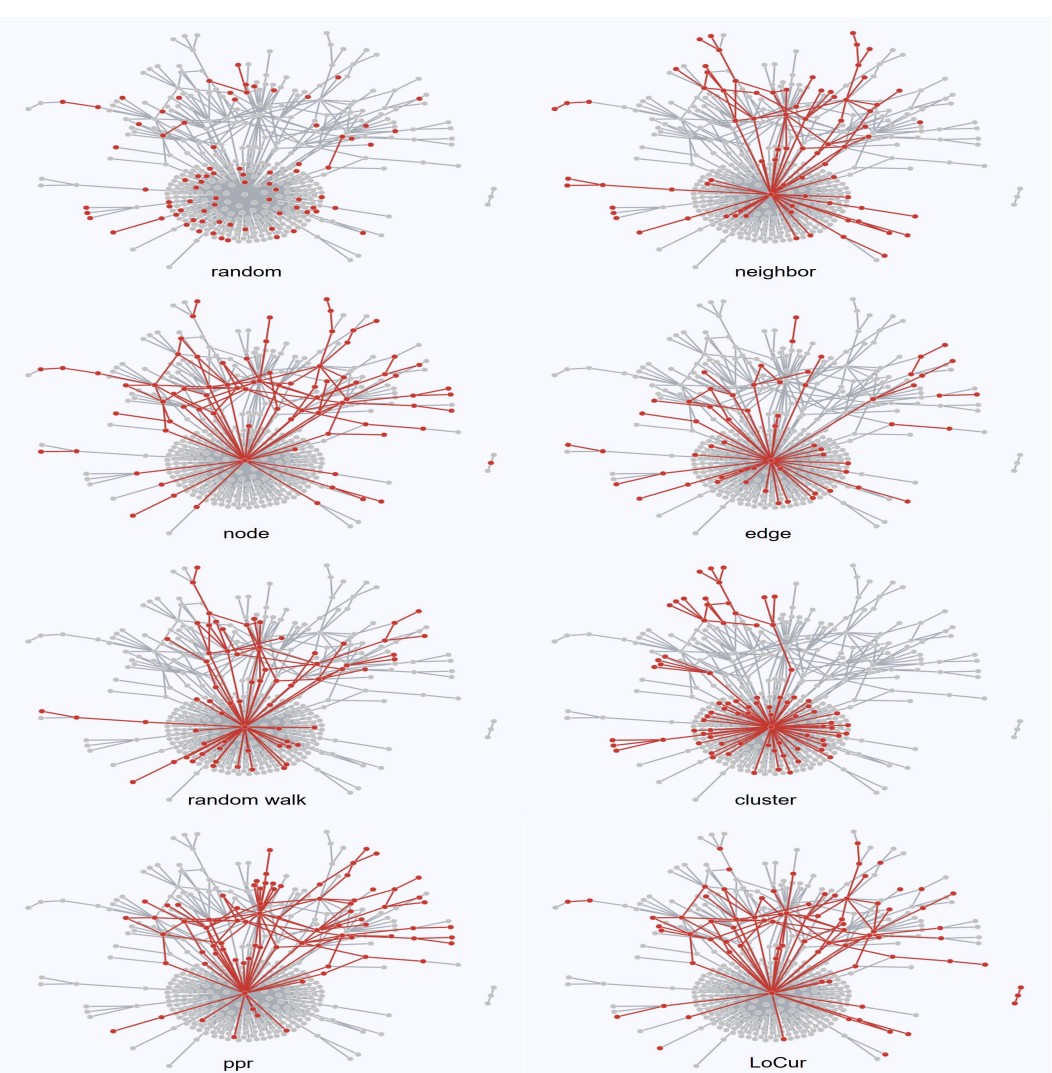

Figure 6: Subgraph examples for the REDDIT-BINARY graph.

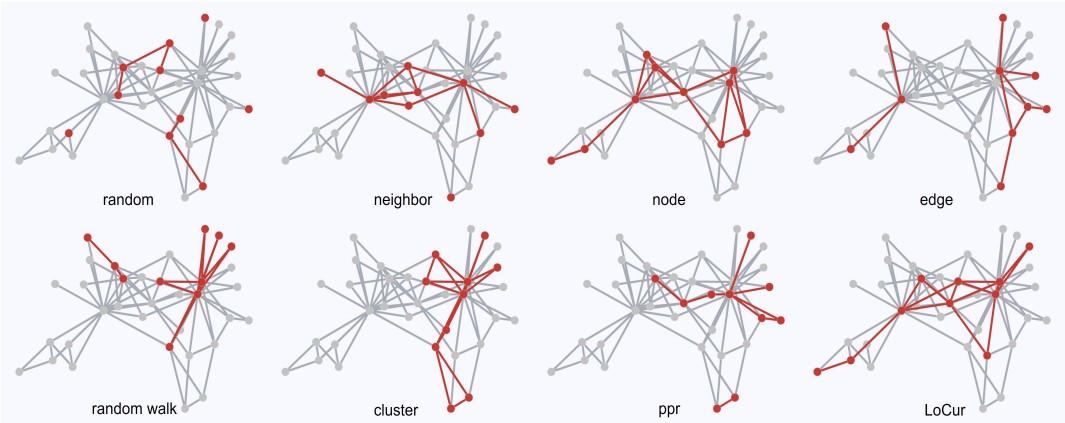

Figure 7: Subgraph examples for the Karate club graph.

# Appendix References

[46] Paul D. Dobson and Andrew J. Doig. Distinguishing enzyme structures from non-enzymes without alignments. *Journal of Molecular Biology*, 330(4):771–783, 2003.

[47] Federico Errica, Marco Podda, Davide Bacciu, and Alessio Micheli. A fair comparison of graph neural networks for graph classification. *arXiv preprint arXiv:1912.09893*, 2019.

[48] Florentin Münch and Radosław K Wojciechowski. Ollivier ricci curvature for general graph laplacians: Heat equation, laplacian comparison, non-explosion and diameter bounds. *Advances in Mathematics*, 356:106759, 2019.

[49] Chien-Chun Ni, Yu-Yao Lin, Jie Gao, Xianfeng David Gu, and Emil Saucan. Ricci curvature of the internet topology. In *2015 IEEE conference on computer communications (INFOCOM)*, pages 2758–2766. IEEE, 2015.

[50] Johan Ugander, Lars Backstrom, and Jon Kleinberg. Subgraph frequencies: Mapping the empirical and extremal geography of large graph collections. In *Proceedings of the 22nd International Conference on World Wide Web*, WWW '13, page 1307–1318, New York, NY, USA, 2013. Association for Computing Machinery.

[51] Jinhuan Wang, Pengtao Chen, Bin Ma, Jiajun Zhou, Zhongyuan Ruan, Guanrong Chen, and Qi Xuan. Sampling subgraph network with application to graph classification. *IEEE Transactions on Network Science and Engineering*, 8(4):3478–3490, 2021.

[52] Melanie Weber, Jürgen Jost, and Emil Saucan. Detecting the coarse geometry of networks. In *NeurIPS Relational Representation Learning*, 2018.

[53] Peter Wills and François G Meyer. Metrics for graph comparison: a practitioner's guide. *Plos one*, 15(2):e0228728, 2020.

[54] Keyulu Xu, Weihua Hu, Jure Leskovec, and Stefanie Jegelka. How powerful are graph neural networks? In *International Conference on Learning Representations*, 2019.

[55] Qi Xuan, Jinhuan Wang, Minghao Zhao, Junkun Yuan, Chenbo Fu, Zhongyuan Ruan, and Guanrong Chen. Subgraph networks with application to structural feature space expansion. *IEEE Transactions on Knowledge and Data Engineering*, 33(6):2776–2789, 2019.

[56] Pinar Yanardag and S. V. N. Vishwanathan. Deep graph kernels. In *KDD*, pages 1365–1374, 2015.