# OpenReview forum: "Localized Curvature-based Combinatorial Subgraph Sampling for Large-scale Graphs"
_NeurIPS.cc/2022/Conference — NeurIPS 2022 Submitted_

### Official Review · Reviewer_oyAc · 2022-07-02

**Rating:** 3
**Confidence:** 4
**Soundness:** 2 fair
**Presentation:** 1 poor
**Contribution:** 2 fair

**Summary:**

The paper introduces an approach to (large) graph subsampling based on positive Ollivier Ricci curvature preservation. This is motivated by theoretical analysis showing that local structure can be better preserved by sampling positively curved edges. Experiments are conducted to support the findings on tasks like node-classification.

**Questions:**

The more technical/theoretical/presentation questions are already listed in the section above.

Further questions:

- How do you compare to the baseline in terms of complexity and time?


**Limitations:**

Limitations have not been addressed.

**Strengths And Weaknesses:**

Originality and impact:

As far as I can tell, the core idea of leveraging directly the Ollivier curvature by relying on positively curved edges for sampling procedures is new. Some of the theoretical findings are in my opinion not that significant though.

- Proposition 1 feels a lot like simply `shifting' the problem with the distance d_{m} that should be defined in terms of random walks Markov chains exactly as for the Ollivier curvature. Similarly, Proposition 3 states something relatively obvious that is for sparse graphs (where higher-order structures are in fact less frequent on average) neglecting higher order structures in the curvature computation would indeed be not that costly.
- Partly connected to the first point, results like Proposition 2 and Proposition 3 feel more like `gap-filling' and less coherent with the narrative.


Quality and soundness:

The presentation is poor and the general soundness is at points questionable. Some instances are listed below (but the actual list is longer):
- Lines 102--104: the message should be better fleshed out here.
- Is Definition 1 novel? This seems like OT distance definition yet there is no reference. Also line 110 `We define... around the nodes' is not clear.
- Where is equation (2) derived?
- The `local structural graph' used throughout the paper $\mathcal{G}_{x}$ is never introduced -- I can only guess what that is.
- I don't quite follow the first inequality in (23), could you elaborate?
- What is $\nabla_{x_{i}y_{i}}$ in (7)?
- The sentence `The gradient is proportional.. in the path' at line 194 is unjustified.
- In the proof of Proposition 2, how (30) derives from (29)? If $m$ is not supported in $y$, then we should have $(D^{-1}Af)(y)$ and not $f(y) + \Delta f (y)$.
- In (31) what is $\nabla_{yx}$?
- In (36) there should not be an equality since we are bounding (32) and (33) from above?
- From the previous points, I am a doubtful about the correctness of Proposition 2.
- I don't follow paragraph 272-277 and similarly the scenario for the node-classification task is not clear (see Questions section as well).
- Appendices are not useful, mostly containing parts copied/reported by existing references and fail to further clarify and explain the paper.

---

### Official Review · Reviewer_mCsD · 2022-07-08

**Rating:** 5
**Confidence:** 3
**Soundness:** 2 fair
**Presentation:** 3 good
**Contribution:** 2 fair

**Summary:**

The paper presents a new method for picking samples consisting of relatively small subgraphs of a very large graph so that the selected subgraphs are representative of the large graph for learning and classification.  It is stated that this process improves tasks involving classification of the graph using the small samples, eg as in GNNs.  The sampling method is based on constructing distributions on nodes and edges of the graph which reflect their discrete Ricci curvatures.  Subsequently, several key observations about the localized curvature-based node/edge distributions are stated and proved. Numerical experiments are provided at the end which support the claim of representativeness of the samples for graph learning and classification.

**Questions:**


It would be helpful if the authors could address each of the questions raised above:

1) Why do sampled subgraphs used in feature learning need to be similar in any way to the larger graph? Would it not be better if these samples cover the spectrum of variations that subgraphs of a fixed size actually exhibit?

2) what actual graph classification tasks did the computational experiments solve?  More specifically, what are the classification/learning problems in ogbn-arxiv and ogbn-mag tasks which represent the problems set out to handle via small subgraph sampling?

3) How does the proposed method compare with prior art?  The authors cite much prior work.  Are the rows labeled random, neighbor, node, edge, rw, cluster and ppr the best known prior sampling methods and if so could the authors remind the reader what classification learning tasks obgn consisted of?

**Strengths And Weaknesses:**


The methodology presented is theory-based and is computationally manageable.   The theoretical results are discussed at some length in the supplementary material provided.

Three key weaknesses of this paper are 1) Why do sampled subgraphs (segments of the very large graph one wishes to learn) used in feature learning need to be similar in any way to the larger graph, the enormous discrepancy between their node/edge sizes notwithstanding, 2) what actual graph classification tasks did the computational experiments solve?  and 3) How does the proposed method compare with prior art?

---

### Official Review · Reviewer_oc4B · 2022-07-08

**Rating:** 4
**Confidence:** 4
**Soundness:** 2 fair
**Presentation:** 2 fair
**Contribution:** 3 good

**Summary:**

This paper introduces a subgraph sampling method based on curvature to train large-scale graphs via mini-batch training. They define a combinatorial metric that distributionally measures the similarity between an original graph and all possible node and edge combinations of the subgraphs. This metric is used as an evaluation metric for how good the sampling is. They found that sampling the edges with large curvatures is equivalent to reducing the distributional difference.

**Questions:**


1. Since the sampling probability p_xy is proportional to k_xy, and k_xy can be negative, how can you get a negative probability?

2. Proposition 3 shows the lower error bound of localized curvature. I am not sure whether this lower bound is small. Since the range of Ollivier Ricci curvature is usually between -1 and 1, the lower error bound between \hat{k_xy} and k_xy should not be greater than 2. Please provide more statements about when the error bound is small, and when it is large, either in a theoretical way or an empirical experiment way.


**Limitations:**

It is not clear what are the drawbacks or challenges of existing network sampling methods and how the authors address them.

**Strengths And Weaknesses:**


pros:
1. The motivation is well explained. It is true that the large graph is computationally infeasible, especially in GCN. Network sampling is in desperate need.
 2. The proposed method for sampling is novel.
 3. The writing and presentation are nice.

cons:
This paper emphasizes too many selling points, such as proposing a  distribution metric, proposing a curvature-based sampling, proposing localized curvature calculation, and improving the accuracy of GCN. But I have two questions. (1) Regarding the distribution metric, I am confused about the innovation of this metric. As for me, this metric is the same as the definition of Gromov-Wasserstein distance. Please clarify the difference. (2) Regarding the localized curvature calculation, which basically uses the lower bound which is derived in [19]. What's the innovation in this paper?

---

### Official Review · Reviewer_ybPg · 2022-07-11

**Rating:** 3
**Confidence:** 4
**Soundness:** 2 fair
**Presentation:** 2 fair
**Contribution:** 2 fair

**Summary:**

The paper proposes a curvature-based graph subsampling method that aims at sampling structurally representative subgraphs via Olliver's Ricci curvature.

**Questions:**

see above

**Limitations:**

see above

**Strengths And Weaknesses:**

*Strength*
The paper addresses an important topic with geometric tools that are not very well explored in this context.

*Weaknesses*
- The writing should be improved. In terms of clarity and presentation, I don't think that the paper in its current form meets the standards of NeurIPS. Examples: theoretical results are stated as propositions in the main text with no indication of whether and where proof can be found. The actual downstream tasks on which you benchmark your algorithm are not described in the main text and there are no clear references to the appendix.
- It is not clear, whether your approach is computationally efficient and how its efficiency compares to the other methods you compare against. This should be reported in addition to a comparison of the achieved accuracy.
- The literature review is not very extensive. Related work on applications of curvature in network analysis is only briefly mentioned in section 2; importantly, related methods on *curvature-based sampling* are only mentioned in the appendix. This should be moved to the main text.
- The proposed theoretical guarantees for the curvature approximation are given for the Erdos-Renyi (ER) model only. This is not necessary a good model for real network data. In addition, the structure of ER graphs varies very significantly with the edge threshold, which is not even commented on. Do your results hold for all regimes?
- What are the values in Tab. 1? How are they computed?

---

### Meta-Review · Area_Chair_zH69 · 2022-08-26

**Recommendation:** Reject
**Confidence:** Certain

**Metareview:**

The majority reviewers consider that this paper should be rejected.  Their concerns include clarity of presentation, a comparison to previous work and finally a number of individual points which were not addressed in the rebuttal period.


**Award:**

No

---

### Decision · Program_Chairs · 2022-09-14

Reject